# Systems consolidation induces multiple memory engrams for a flexible recall strategy in observational fear memory in male mice

Joseph I. Terranova [1,2], Jun Yokose[1], Hisayuki Osanai [1], Sachie K. Ogawa[1] & Takashi Kitamura [1,3] ✉

Observers learn to fear the context in which they witnessed a demonstrator's aversive experience, called observational contextual fear conditioning (CFC). The neural mechanisms governing whether recall of the observational CFC memory occurs from the observer's own or from the demonstrator's point of view remain unclear. Here, we show in male mice that recent observational CFC memory is recalled in the observer's context only, but remote memory is recalled in both observer and demonstrator contexts. Recall of recent memory in the observer's context requires dorsal hippocampus activity, while recall of remote memory in both contexts requires the medial prefrontal cortex (mPFC)-basolateral amygdala pathway. Although mPFC neurons activated by observational CFC are involved in remote recall in both contexts, distinct mPFC subpopulations regulate remote recall in each context. Our data provide insights into a flexible recall strategy and the functional reorganization of circuits and memory engram cells underlying observational CFC memory.

Animals can directly learn to associate aversive stimuli and the cues that predict danger. However, direct experience of aversive stimuli and situations is harmful to individuals and high-risk because, in some cases, these are lethal. Thus, there is strong evolutionary pressure that favors vicarious learning in different animal species, including zebrafishes, cows, cats, rodents, non-human primates, and humans[1–7]. Observational fear is an empathic response in which an observer witnesses a demonstrator in a dangerous situation and responds with fear behavior[5,8,9]. Observational fear facilitates vicarious associative learning in the observer, allowing the observer to ascertain dangerous stimuli and situations by witnessing the demonstrator's aversive moments[10–15]. Previous studies have demonstrated that observers remember the context in which they witnessed aversive stimuli delivered to the demonstrator[5,12–18]. We refer to this as observational contextual fear conditioning (observational CFC). The anterior cingulate

cortex (ACC) to amygdala circuit is essential for acquisition of observational CFC memory[16,19]. A subset of ACC neurons in the observer respond to aversive events occurring in the demonstrator, which delivers an unconditioned stimulus (US) to the amygdala and the US signal is associated with the context as a conditioned stimulus (CS) to form observational CFC memory in the amygdala[5,16,18,19].

However, several outstanding questions regarding both behavioral and neurobiological aspects of observational CFC memory remain unexplored. First, how long does observational CFC memory persist in the observer? Observational CFC memory is immensely adaptive because it allows the observer to learn about a dangerous context without having direct aversive experience. Therefore, it would be reasonable to expect that observational CFC memory would persist for days or weeks after initial observational CFC experience. While memory of direct CFC experience is present at recent (1 day) and

[1]Department of Psychiatry, University of Texas Southwestern Medical Center, Dallas, TX 75390, USA. [2]Department of Anatomy, Midwestern University, Downers Grove, IL 60615, USA. [3]Department of Neuroscience, University of Texas Southwestern Medical Center, Dallas, TX 75390, USA. ✉e-mail: Takashi.Kitamura@UTSouthwestern.edu

remote (30 days and even longer) time points[20-30], the duration of observational CFC memory has not been characterized.

Second, can observational CFC memory be recalled in observer's and demonstrator's context? Recall of observational CFC memory from observer's point of view would mean that observers recall the specific details of their own fear experience while they witnessed aversive stimuli delivered to the demonstrator[31,32]. On the other hand, recall of observational CFC memory in demonstrator's point of view would imply that observers have a conceptual understanding of the demonstrator's aversive experience and can therefore extrapolate that their own fear experience is linked to the demonstrator's chamber, thus allowing observers to predict that the aversive events may occur to themselves[31,32]. Flexible use of recall strategies in observational CFC memory is highly advantageous because it would enable the observer to avoid potential dangers from different standpoints. Previous studies reported that observers can recall observational CFC memory in the observer chamber but not in the demonstrator chamber at the recent

time point[5,12-18,33]. Since memory of direct experience transforms from episodic to semantic over time in humans[34-36] and direct CFC memory generalizes over time in rodents[37-47], if observational CFC memory is long-lasting, it is possible that the quality and/or recall strategy of observational CFC memory changes over time to enable the recall in the demonstrator chamber. However, it has not been considered whether observational CFC memory can be recalled at the remote time point.

Finally, what are the neural circuit mechanisms necessary for recall of observational CFC memory at recent and remote time points? Recent recall of direct CFC memory requires a hippocampal (HPC)-amygdala circuit as dorsal hippocampus responds to the conditioned context that elicits the recall of associative fear memory formed in basolateral amygdala (BLA), whereas remote recall of direct CFC memory requires a medial prefrontal cortex (mPFC)-BLA circuit since mPFC neurons are activated by the exposure to the conditioned context at the remote time point, instead of hippocampus[20,21,46-56]. This

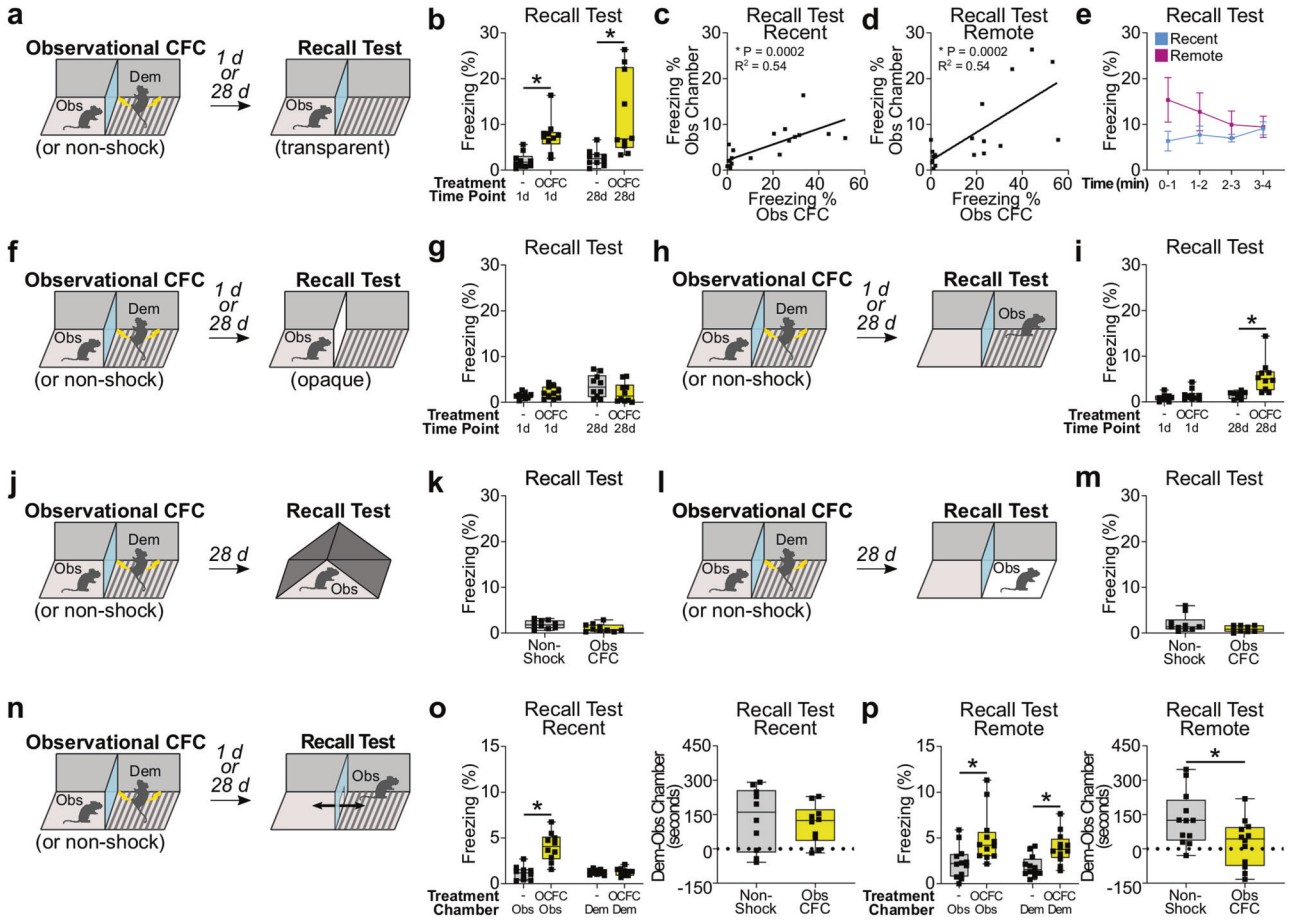

**Fig. 1 | Recall of observational contextual fear conditioning memory (observational CFC; Obs CFC; OCFC).** Obs observer, Dem demonstrator, d, Day. **a, b** Recall of observational CFC memory in observer chamber with transparent partition. −; non-shock group. **c, d** Correlation of observer freezing levels during observational CFC with freezing levels during recall of observational CFC memory in observer chamber at the recent (**c**) or remote (**d**) time points. **e** Time course of observer freezing levels during recall of observational CFC memory in observer chamber at the recent or remote time points. min; minute. **f, g** Recall of observational CFC memory in observer chamber with opaque partition. **h, i** Recall of observational CFC memory in demonstrator chamber. **j, k** Recall of observational CFC memory in a novel context. **l, m** Recall of observational CFC memory in a novel context that preserves the view of the observer chamber. **n** Chamber preference after observational CFC. **o, p** (Left) Observer freezing levels in the observer or demonstrator chambers at the recent (**o**) or remote (**p**) time point. (Right) Effect of

observational CFC on chamber preference at recent (**o**) or remote (**p**) time points. Dotted line; equal chamber preference (0 s). Two-way between-subjects ANOVA with Bonferroni test (**b, g, i, o, p**), two-tailed Pearson's correlation coefficient (**c, d**), two-way mixed ANOVA (**e**), unpaired $t$ tests (**k, o, p**), and Mann−Whitney $U$-test (**m**) were performed. *$P < 0.05$, bars without asterisks did not reach significance ($P > 0.05$). Graphs are presented as boxplots with minimum (lower whisker), 25th percentile (lower box bound), median (center), 75th percentile (upper box bound), and maximum (upper whisker) values indicated, except for (**c, d**), which are presented as correlations, and (**e**), which is presented as line graphs with data presented as mean values +/− SEM. Gray bar; non-shock group. Yellow bar; observational CFC group. For a complete description of statistics for this and all subsequent figures, please see Supplemental Table 1. Source data are provided as a Source data file.

neural circuit-reorganization process is known as systems consolidation of memory[57–63], which is a common process observed in several different mammalian species (mice, rats, rabbits, cats, monkeys, and humans)[26,29,61,64–66]. Systems consolidation of direct CFC memory is supported by the rapid generation of engram cells in the mPFC, which are defined as a subpopulation of neurons that undergo biophysical changes to encode a specific memory episode[67–70]. Over time, these fear memory engram cells within the mPFC gradually mature, thus enabling them to regulate the recall of direct CFC memory at the remote time point[21,46,52,53,63,69,71–73]. However, it remains unknown whether there are similar neural circuit mechanisms for observational CFC memory or if these mechanisms are totally different. Relatedly, if observers can recall observational CFC memory in both the observer and demonstrator chambers, are the underlying neural circuit mechanisms similar or different? Based on the differential perspective of observers during recall of observational CFC memory in the observer and demonstrator chambers, we speculate that the corresponding neural circuit mechanisms would also be different.

In this study, we subjected mice to the memory recall test for observational CFC at the recent (1 day after observational CFC) or remote (28 days after observational CFC) time points, and then examined the neural circuit mechanisms that regulate the recall of observational CFC memory. We found that observational CFC memory is long-lastingly maintained. Although recall of observational CFC memory at the recent time point only occurs in the observer chamber, recall of observational CFC memory at the remote time point occurs in both the observer and demonstrator chambers. We identified that recall of observational CFC memory at the recent time point requires dorsal HPC (dHPC) while, surprisingly, recall of observational CFC memory in both chambers at the remote time point requires the mPFC-BLA pathway. Next, we demonstrated that activation of a subset of mPFC neurons during observational CFC is essential for the remote memory formation of observational CFC, and that the subpopulation of mPFC neurons activated by observational CFC is reactivated during recall of observational CFC memory in both the observer and demonstrator chambers at the remote time point. Finally, we demonstrated that there are distinct subpopulations of mPFC neurons that are associated with recall of observational CFC memory in either the observer or demonstrator chamber at the remote time point. Therefore, we propose that systems consolidation of observational CFC memory generates a new subpopulation of observational CFC memory engram cells in mPFC, which enables observers to engage in a flexible recall strategies of observational CFC memory at the remote time point.

## Results
### Recall of observational CFC memory in the observer and demonstrator chambers at the recent and remote time points
We subjected observers to observational CFC (see "Methods") and, 1 or 28 days later, we examined recall of observational CFC memory by reintroducing observers to the conditioned context from which they were subjected to observational CFC (i.e., the observer chamber) (Fig. 1a). We found a significant main effect of observational CFC compared with the non-shock group on recall of observational CFC memory in the observer chamber, such that observational CFC induced the observer freezing response in the observer chamber at the recent time point, which is consistent with previous studies[5,12–18], and at the remote time point, which had not yet been examined (Fig. 1b). Recall of observational CFC memory in the observer chamber at the recent and remote time points positively correlates with observer freezing levels during observational CFC, and there was no change in the kinetics of observer freezing levels throughout the duration of the recall test at either time point (Fig. 1c–e). Next, we examined if the view of the demonstrator chamber from the observer chamber serves as the conditioned stimulus (CS) to trigger recall of observational CFC

memory in the observer chamber by using an opaque partition, which we predicted would suppress the observer freezing response (Fig. 1f). We found that, when the view of the demonstrator chamber was blocked by an opaque partition, we were unable to detect a difference in freezing level between observers subjected to observational CFC and the non-shock group (Fig. 1g). Consistent with these results, observer head direction during freezing is preferentially oriented towards the demonstrator chamber at both time points, suggesting that the view of the demonstrator chamber is an important cue to trigger the observer's freezing response (Supplemental Fig. 1a, b). These results indicate that recall of observational CFC memory in the observer chamber is CS-dependent and long-lasting.

Next, we examined whether observational CFC memory can be recalled in the demonstrator chamber at the recent and remote time points. While a previous study suggests that observers may not show a freezing response in the demonstrator chamber at the recent time point[33], it remains unknown whether they can recall observational CFC memory in the demonstrator chamber at the remote time point. To address this question, we subjected observers to observational CFC, and then tested recall of observational CFC memory in the demonstrator chamber at the recent and remote time points (Fig. 1h). There was a significant interaction between observational CFC and time point in which, at the recent time point, observers subjected to recall of observational CFC memory in the demonstrator chamber had similar freezing levels compared with the non-shock group, whereas, at the remote time point, observers subjected to recall of observational CFC memory in the demonstrator chamber had higher freezing levels compared with the non-shock group (Fig. 1i). There was no correlation between observer freezing levels during observational CFC with recall of observational CFC memory in the demonstrator chamber at the recent or remote time points, and there was no change in the kinetics of observer freezing levels throughout the duration of the recall test at either time point (Supplemental Fig. 2a–c). Moreover, the head direction of observers is not preferentially oriented during recall of observational CFC memory in the demonstrator chamber at the remote time point (Supplemental Fig. 1c, d). These data suggest that, over time, observers gain the ability to recall observational CFC memory in the demonstrator chamber.

To determine if observational CFC memory generalizes over time, we subjected observers to observational CFC and then introduced them to a novel context at the remote time point (Fig. 1j). Although we predicted that observational CFC would increase observer freezing levels to a novel context at the remote time point compared with the non-shock group, which would indicate generalization of observational CFC memory, we could not find that observational CFC increases observer freezing levels (Fig. 1k). To further examine potential generalization of observational CFC memory, we subjected observers to observational CFC and then introduced them to a new context that was the same as the demonstrator chamber except for a plastic opaque floor that covered the shock grid (Fig. 1l). While we predicted that observational CFC would increase observer freezing levels compared with the non-shock group in this context that is very similar to the demonstrator chamber, again we could not find that observational CFC increases observer freezing levels (Fig. 1m). Together, these data suggest that the elevated freezing levels by observers in the demonstrator chamber cannot be explained by memory generalization.

Finally, to directly demonstrate if observers specifically develop fear for the demonstrator chamber at remote time point, we subjected observers to observational CFC and then tested them for their preference of the observer or demonstrator chamber at the recent and remote time points. We introduced observers to a modified chamber that contained an opening in the center of the transparent partition (Fig. 1n), and examined their freezing response in the observer or demonstrator chambers at the recent or remote time

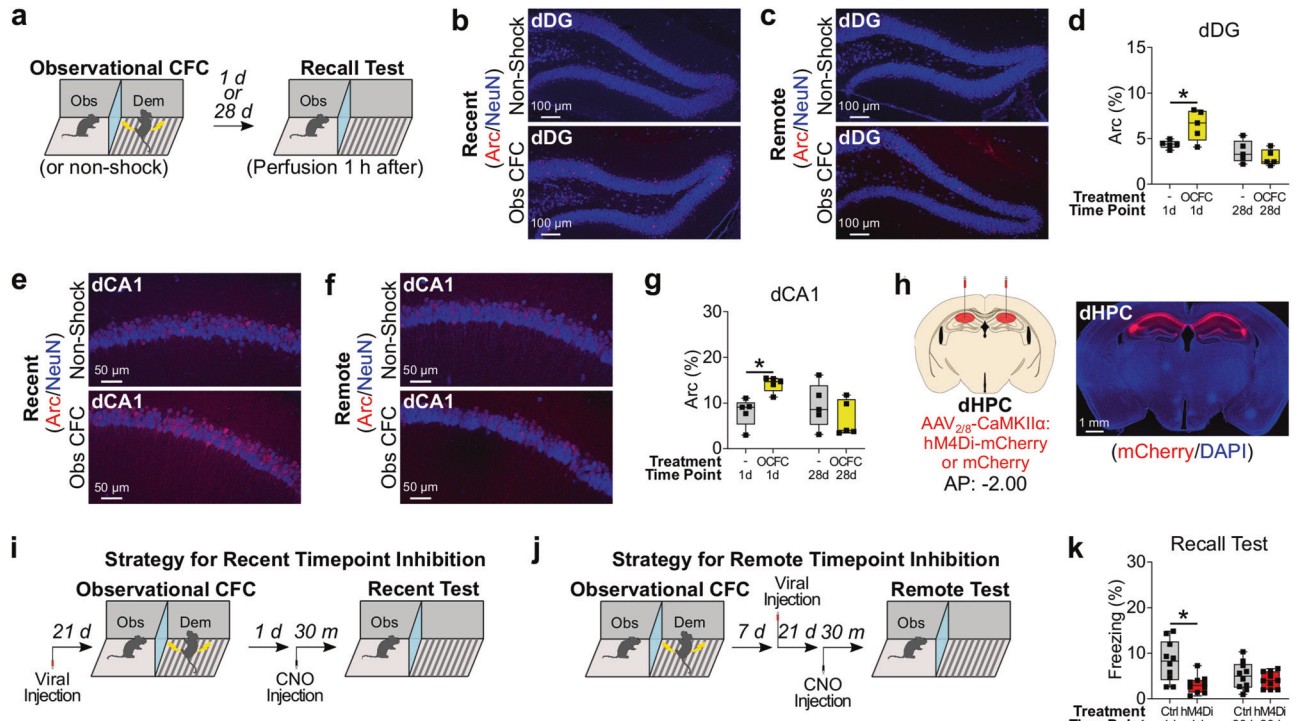

**Fig. 2 | Role of dorsal hippocampus (dHPC) in recall of observational CFC memory in observer chamber. a** Schedule. h hour. **b**, **c** Coronal sections of dorsal Dentate Gyrus (dDG) immunohistochemistry in recall of observational CFC memory at the recent (**b**) or remote (**c**) time point. Top: Non-Shock group. Bottom: Obs CFC group. **d** Percentage of Arc+ neurons in dDG in recall of observational CFC memory. **e**, **f** Coronal sections of dorsal CA1 (dCA1) immunohistochemistry in recall of observational CFC memory at the recent (**e**) or remote (**f**) time point. Top: Non-Shock group. Bottom: Obs CFC group. **g** Percentage of Arc+ neurons in dCA1 in recall of observational CFC memory. **h** (Left) Injection strategy. AP anteroposterior

coordinate. (Right) Coronal section of hM4Di-mCherry expression in dHPC. **i**, **j** Schedule for chemogenetic inhibition of dHPC at the recent (**i**) or remote (**j**) timepoint. **k** Effect of chemogenetic inhibition of dHPC on recall of observational CFC memory. Ctrl; Control. Gray bar; Ctrl group. Red Bar; hM4Di group. Two-way between subjects ANOVA with Bonferroni test (**d**, **g**, **k**) was performed. *$P < 0.05$, and bars without asterisks did not reach significance ($P > 0.05$). Graphs are presented as box plots with minimum (lower whisker), 25th percentile (lower box bound), median (center), 75th percentile (upper box bound), and maximum (upper whisker) values indicated. Source data are provided as a Source data file.

points (Fig. 1o, p). At the recent time point, observers subjected to observational CFC had significantly higher freezing levels in the observer chamber but not demonstrator chamber compared with the non-shock group, and we were unable to detect a difference in chamber preference between observers in both groups (Fig. 1o). In contrast, at the remote time point, we found that observers in the observational CFC group had significantly higher freezing levels in both the observer and demonstrator chambers compared with the non-shock group, and observers in the observational CFC group spent less time in the demonstrator chamber compared with the non-shock group (Fig. 1p). Together, these findings indicate that, at the recent time point, observers engage in recall of observational CFC memory from their perspective. In contrast, at the remote time point, observers subjected to observational CFC learn to fear the demonstrator chamber. Thus, at the remote time point, observers can adopt a flexible recall strategy for observational CFC memory.

### Role of dorsal hippocampus (dHPC) in recall of observational CFC memory in the observer chamber
Since dHPC is essential for the formation and recall of contextual memory for direct CFC at the recent time point, we examined the role of dHPC activity on recall of observational CFC memory in the observer chamber at the recent and remote time points. We subjected observers to observational CFC and then quantified expression of the immediate early gene, Arc (activity-regulated cytoskeleton-associated protein)[74–76], as a marker of neural activity during recall of observational CFC memory in the observer chamber at the recent or remote time points in dHPC subregions dorsal dentate gyrus (dDG) and dorsal CA1 (dCA1) (Fig. 2a–g)[74,75]. There was a significant interaction between

observational CFC exposure and time point, where Arc positivity was significantly higher in both dDG and dCA1 in the observational CFC group compared with the non-shock group at the recent time point but not the remote time point (Fig. 2d, g). These data suggest that dHPC activity may be essential for recall of observational CFC memory in the observer chamber at the recent but not remote time point. To test the necessity of dHPC activity, we performed chemogenetic neural silencing of dHPC neurons by injecting adeno-associated virus 2/8 (AAV2/8)-CaMKII:hM4Di-mCherry or AAV2/8-CaMKII:mCherry as a control into dHPC bilaterally (Fig. 2h). We subjected observers to observational CFC, and then inhibited dHPC during recall of observational CFC memory in the observer chamber at the recent time point or remote time point by injecting Clozapine N-oxide (CNO) 30 min before the recall test (Fig. 2i, j). Chemogenetic inhibition of dHPC activity impaired observer freezing levels in recall of observational CFC memory in the observer chamber at the recent but not remote time point (Fig. 2k). Therefore, the dHPC activity is necessary for recall of observational CFC memory in the observer chamber at the recent time point.

### Role of mPFC in recall of observational CFC memory in observer chamber
To examine the role of mPFC activity on recall of observational CFC memory in the observer chamber, we quantified Arc expression in the following mPFC subregions during recall of observational CFC memory at the recent or remote time points: the secondary motor area (SOM), dorsal part of the anterior cingulate area (ACAd), prelimbic cortex (PLC), and infralimbic cortex (ILC) (Fig. 3a, b). There were significant interactions between observational CFC and time point on Arc

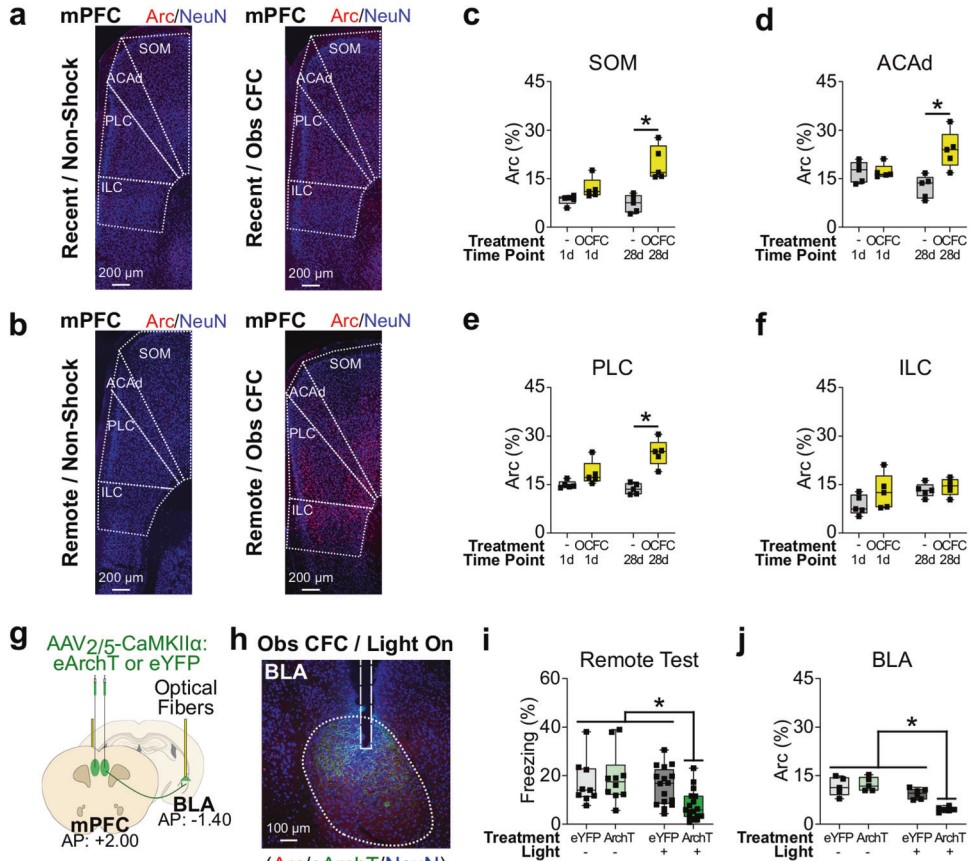

**Fig. 3 | Role of medial prefrontal cortex (mPFC) in recall of observational CFC memory in the observer chamber. a, b** Coronal section of mPFC immunohistochemistry in non-shock (left) or recall of observational CFC memory (right) at the recent (**a**) or remote (**b**) time point. SOM secondary motor area, ACAd, anterior cingulate area (dorsal subregion), PLC prelimbic cortex, ILC infralimbic cortex. **c–f** Percentages of Arc+ neurons in mPFC subregions SOM (**c**), ACAd (**d**), PLC (**e**), and ILC (**f**) in recall of observational CFC memory. **g** Effect of optogenetic inhibition of mPFC terminals in BLA in recall of observational CFC memory in observer chamber at the remote timepoint. **h** Coronal section of BLA immunohistochemistry after mPFC terminal inhibition in recall of observational CFC memory in observer chamber. **i, j** Effect of optogenetic terminal inhibition on recall of observational CFC memory in observer freezing levels (**i**) and percentages of Arc+ neurons in BLA (**j**). –; Light off. +; Light on. Light gray bar; eYFP/Light off. Light green bar; eArchT/Light off. Dark gray bar; eYFP/Light on. Dark green bar; eArchT/Light on. Two-way mixed ANOVA with Bonferroni test (**c–f**) and two-way between subjects ANOVA with Tukey test (**i–j**) were performed. *$P < 0.05$, and bars without asterisks did not reach significance ($P > 0.05$). Graphs are presented as box plots with minimum (lower whisker), 25th percentile (lower box bound), median (center), 75th percentile (upper box bound), and maximum (upper whisker) values indicated. Source data are provided as a Source data file.

positivity in the SOM, ACAd, and PLC during recall of observational CFC memory in the observer chamber, such that Arc positivity was significantly higher in the observational CFC group compared with the non-shock group at the remote time point but not recent time point (Fig. 3c–e). For ILC, we did not detect a main effect of observational CFC, time point, or interaction between observational CFC and time point on Arc positivity (Fig. 3f). Together, these data suggest that the mPFC, specifically SOM, ACAd, and PLC subregions may be essential for recall of observational CFC memory in the observer chamber at the remote time point but not recent time point. Since BLA is necessary for recent recall of observational CFC memory[5,16,18], we examined the necessity of the mPFC-BLA pathway during recall of observational CFC memory in the observer chamber at the remote time point by performing optogenetic inhibition of mPFC terminals in BLA. We bilaterally injected AAV$_{2/5}$-CaMKIIα:eArchT-eYFP or AAV$_{2/5}$-CaMKIIα:eYFP as a control in mPFC, implanted optical fibers targeting BLA, and then inhibited mPFC terminals in BLA during recall of observational CFC memory in the observer chamber at the remote time point (Fig. 3g, h). Optogenetic inhibition of mPFC terminals in BLA during recall of observational CFC memory in the observer chamber at the remote time point reduced observer freezing level and Arc positivity in BLA (Fig. 3i, j and Supplemental Fig. 3). Together, these data demonstrate

that the mPFC-BLA pathway is necessary for recall of observational CFC memory in the observer chamber at the remote time point.

## Role of dHPC and mPFC in recall of observational CFC memory in the demonstrator chamber

Given the role of dHPC and mPFC activity in recall of observational CFC memory in the observer chamber (Figs. 2 and 3), we investigated dHPC and mPFC activity in recall of observational CFC memory in the demonstrator chamber (Fig. 4a). We quantified Arc expression in dDG (Fig. 4b), dCA1 (Fig. 4c), and mPFC subregions (Fig. 4d–g) after recall of observational CFC memory in the demonstrator chamber at the recent or remote time points. There was no difference in Arc positivity between the observational CFC and the non-shock group at the recent or remote time point in dDG, dCA1, SOM, ACAd, or ILC (Fig. 4b–e, g). In PLC, Arc positivity was higher in the observational CFC group compared with the non-shock group at the remote time point but not at the recent time point (Fig. 4f). These data suggest that the mPFC, specifically the PLC subregion, regulates recall of observational CFC memory in the demonstrator chamber at the remote time point. Therefore, we examined the role of mPFC-BLA pathway in recall of observational CFC memory in the demonstrator chamber at the remote time point (Fig. 4h, i). Optogenetic inhibition of mPFC terminals in BLA also

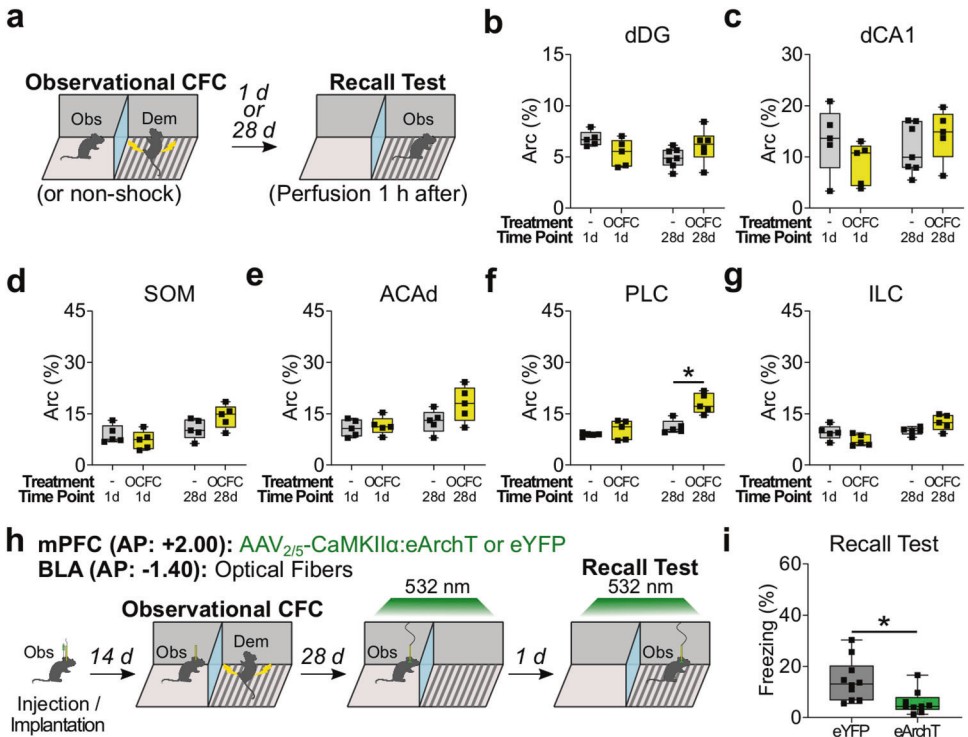

**Fig. 4 | Role of dHPC and mPFC in recall of observational CFC memory in demonstrator chamber. a** Schedule. **b–g** Percentages of Arc+ neurons in dHPC subregions dDG (**b**) and dCA1 (**c**), and in mPFC subregions SOM (**d**), ACAd (**e**), PLC (**f**), and ILC (**g**) in recall of observational CFC memory. **h** Schedule for optogenetic inhibition of recall of observational CFC memory in demonstrator chamber. **i** Effect of optogenetic inhibition of recall of observational CFC memory in demonstrator

chamber. Two-way between subjects ANOVA with Bonferroni test (**b–g**) and unpaired t test (**i**) were performed. *$P < 0.05$, and bars without asterisks did not reach significance ($P > 0.05$). Graphs are presented as box plots with minimum (lower whisker), 25th percentile (lower box bound), median (center), 75th percentile (upper box bound), and maximum (upper whisker) values indicated. Source data are provided as a Source data file.

reduced observer freezing levels during recall of observational CFC memory in the demonstrator chamber at the remote time point (Fig. 4i). These indicate that the mPFC-BLA pathway is necessary for recall of observational CFC memory at the remote time point.

### Role of mPFC neural activity during observational CFC on remote memory formation of observational CFC

Since mPFC neural activity during acquisition of direct CFC memory is necessary for the formation of remote memory of direct CFC memory without affecting recent memory formation, called cortical early tagging[21,63,72,77], we examined whether a similar neural mechanism is involved in the remote memory formation of observational CFC memory, which would enable recall of observational CFC memory in either chamber at the remote time point. We subjected observers to observational CFC and quantified Arc expression in mPFC subregions (Fig. 5a–e). Arc positivity was significantly higher in the observational CFC group compared with the non-shock group in the ACAd and PLC (Fig. 5c, d), and we did not detect a difference in the SOM and ILC (Fig. 5b, e). These suggest that early cortical tagging in mPFC may occur during observational CFC. To test this possibility, we performed chemogenetic neural silencing of mPFC in observers by injecting AAV$_{2/8}$-CaMKII:hM4Di-mCherry or AAV$_{2/8}$-CaMKII:mCherry as a control (Fig. 5f, g). We chemogenetically inhibited mPFC activity during observational CFC by injecting CNO 30 min before observational CFC, and then subjected observers to recall of observational CFC memory in the observer or demonstrator chamber at both the recent and remote time points (Fig. 5h, i). For both chambers, there was a significant interaction between treatment and time point, where chemogenetic inhibition of mPFC activity during observational CFC reduced observer

freezing levels at the remote time point but not recent time point (Fig. 5j, k). These data indicate that the activation of a subset of mPFC neurons during observational CFC is essential for the remote memory formation of observational CFC, which allows observers to recall observational CFC memory in the observer or demonstrator chamber.

Fear memory engram cells in mPFC, which are defined as which are defined as a subpopulation of neurons that undergo biophysical changes to encode a specific memory episode[67–70], regulate the remote recall of direct CFC memory[21,46,52,53,63,69,71–73]. Given the similarities between mPFC in the remote recall of observational CFC memory and direct CFC memory, we considered whether fear memory engram cells associated with the context in mPFC are generated during observational CFC and are reactivated during recall of observational CFC memory in the observer or demonstrator chamber at the remote time point. To examine this question, we injected a cocktail of AAV$_{2/8}$-hSyn-DIO-HA-hM4Di-mCitrine and AAV$_{2/9}$-PRAM:d2tTA-TRE:NLS-mKate2 into the mPFC of TRE-Cre transgenic mice that express Cre recombinase under the control of a tetracycline-responsive promoter element (TRE), and then performed activity-dependent cell tagging to selectively label mPFC neurons activated by observational CFC using the doxycycline-off (off-Dox) Robust Activity Marking (RAM) system. The RAM system directly combines the human c-fos minimal promoter with four tandem repeats of an enhancer module (PRAM) to the destabilized tetracycline transactivator (d2tTA)[78], which binds to TRE, drives the expression of CRE recombinase, and thus allows for the long-term expression of HA-hM4Di-mCitrine in observational CFC-activated neurons (Fig. 5l, m, o). Neurons activated by observational CFC were labeled with mCitrine, and neurons activated by remote recall in the observer chamber (Fig. 5m, n) or the demonstrator chamber (Fig. 5o, p) were labeled with Arc antibodies. We did not

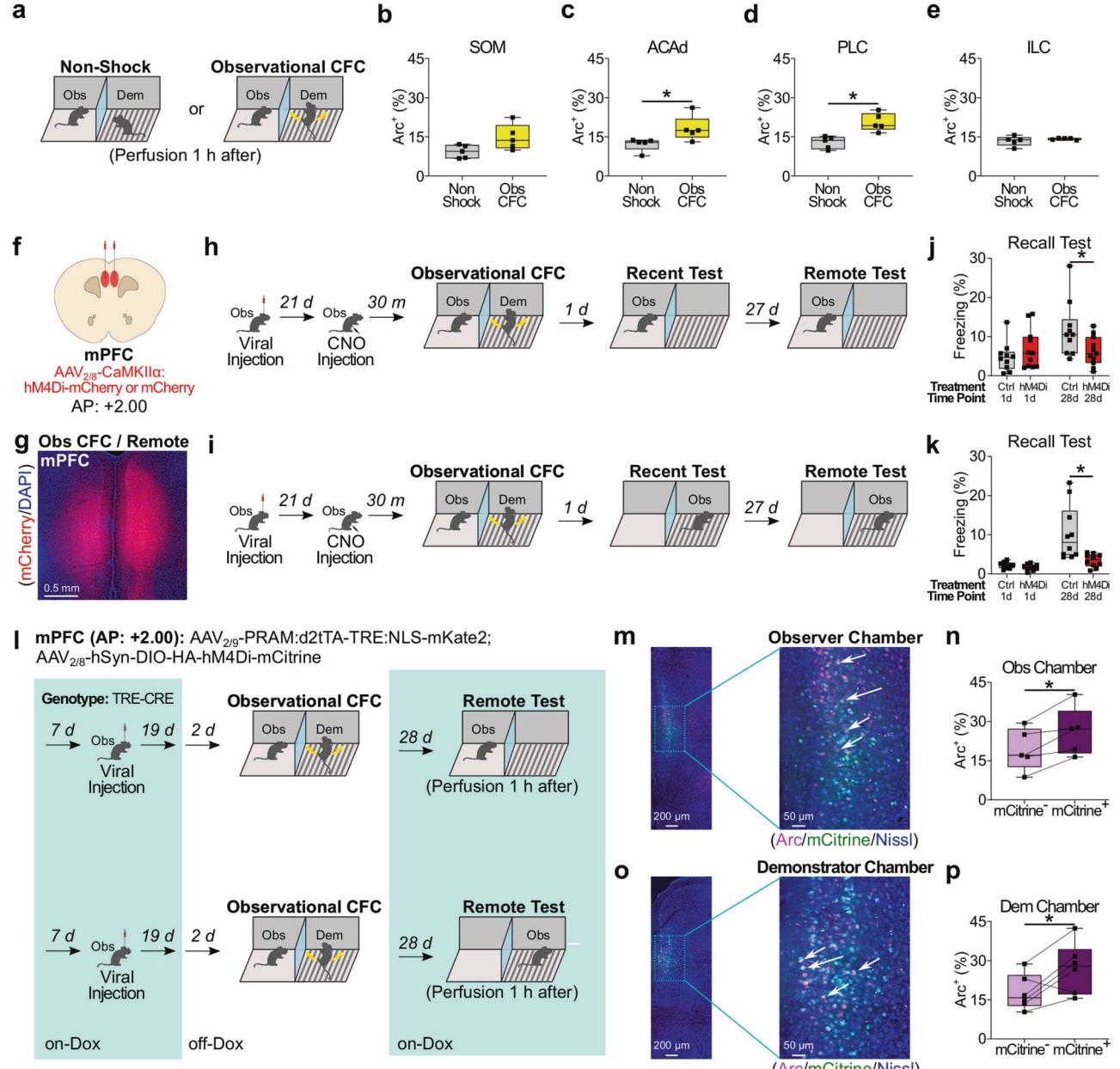

**Fig. 5 | Role of mPFC observational CFC engram cells in recall of observational CFC memory. a** Schedule. **b–e** Percentages of Arc+ neurons in mPFC subregions SOM (**b**), ACAd (**c**), PLC (**d**), and ILC (**e**) in observational CFC. **f** Injection strategy. **g** Coronal section of hM4Di-mCherry expression in mPFC. **h, i** Schedule for effect of mPFC inhibition during observational CFC on observational CFC memory recall in observer (**h**) or demonstrator (**i**) chamber. **j, k** Effect of mPFC inhibition during observational CFC on observational CFC memory recall in observer (**j**) or demonstrator (**k**) chamber. **l** Strategy to label mPFC engram cells with mCitrine. Dox; Doxycycline diet. **m, o** Coronal sections of mPFC engram cell labeling after recall of observational CFC in the observer (**m**) or demonstrator (**o**) chamber. Arrows; mCitrine+Arc+ cells. **n, p** Percentages of Arc+ neurons in mCitrine− or mCitrine+ neurons during recall of observational CFC memory in observer (**n**) or demonstrator (**p**) chamber. Unpaired t test (**b–d**), Mann Whitney U-Test (**e**), two-way mixed ANOVA with Bonferroni test (**j, k**) and paired t test (**n, p**) were performed. *$P < 0.05$, and bars without asterisks did not reach significance ($P > 0.05$). Graphs are presented as box plots with minimum (lower whisker), 25th percentile (lower box bound), median (center), 75th percentile (upper box bound), and maximum (upper whisker) values indicated. Source data are provided as a Source data file.

detect mKate2+ cells, which we attribute to degradation of the mKate2 signal due to the long-term on-Dox condition. We observed that Arc positivity was higher in mCitrine+ neurons compared with the mCitrine− neurons when mice recalled remote fear memory in either chamber (Fig. 5n, p and Supplemental Fig. 4a, b). indicating that the subpopulation of mPFC neurons activated during observational CFC are reactivated during recall of observational CFC memory in the observer or demonstrator chambers at the remote time point. Together, these data suggest that a fear memory engram is generated in mPFC during observational CFC, which is subsequently reactivated to

facilitate recall of observational CFC memory at the remote time point in either chamber.

### Role of mPFC neural subpopulations in the regulation of the recall observational CFC memory in the observer or demonstrator chamber at the remote time point

Our data indicate that similar neural mechanisms regulate recall of observational CFC memory in the observer or demonstrator chamber at the remote time point. However, since recall of observational CFC memory in the observer or demonstrator chamber should be

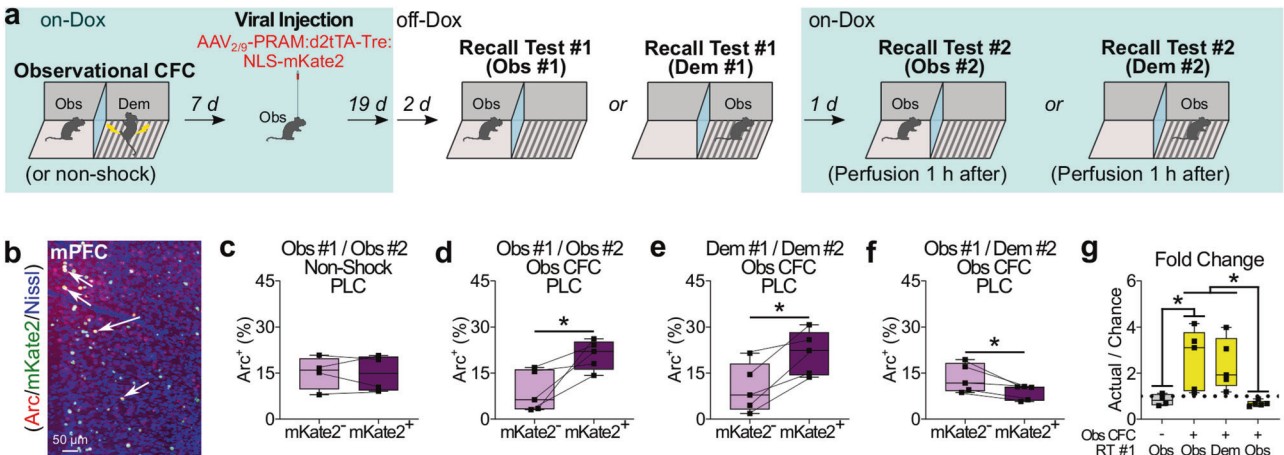

**Fig. 6 | mPFC engram cells activated by recall of observational CFC memory at the remote time point in the observer or demonstrator chamber. a** Strategy to label mPFC engram cells activated in the observer or demonstrator chambers. 28 days after observational CFC, observers were subjected to recall of observational CFC memory in the observer chamber (Obs #1) or demonstrator chamber (Dem #1), which labeled mPFC with mKate2. 1 day later, observers were subjected to a second recall test in either the observer chamber (Obs #2) or demonstrator chamber (Dem #2), which labeled mPFC with Arc. **b** Coronal section of mPFC engram cell labeling after recall of observational CFC memory. mKate2: Recall Test #1. Arc: Recall Test #2. Arrows; mKate2⁺Arc⁺ cells. **c–f** Percentages of Arc⁺ neurons in mKate2⁻ and mKate2⁺ neurons in PLC in Obs #1/Obs #2 (non-shock) (**c**), Obs #1/Obs #2 (Obs CFC) (**d**), Dem #1/Dem #2 (Obs CFC) (**e**), and Obs #1/Dem #2 (Obs CFC) (**f**). **g** Fold change analysis (actual/chance) comparing each behavioral condition. Dotted line; chance level (Fold change = 1). RT; Recall Test. Paired $t$ test (**c–f**), and one-way ANOVA with Tukey test (**g**) were performed. *$P < 0.05$, and bars without asterisks did not reach significance ($P > 0.05$). Graphs are presented as box plots with minimum (lower whisker), 25th percentile (lower box bound), median (center), 75th percentile (upper box bound), and maximum (upper whisker) values indicated. Source data are provided as a Source data file.

fundamentally different processes, we examined whether there are distinct neural subpopulations in mPFC that regulate recall of observational CFC memory in the observer or demonstrator chamber. We injected mice with AAV₂/₉-PRAM:d2tTA-TRE:NLS-mKate2 into mPFC 7 days after observational CFC under the on-DOX condition (Fig. 6a). 28 days after observational CFC, we subjected mice to remote memory test in either the observer or demonstrator chamber under an off-Dox condition and then next day again to the observer or demonstrator chamber under an on-Dox condition (Fig. 6a). Neurons activated by Recall Test #1 were labeled with mKate2, and neurons activated by Recall Test #2 were labeled with Arc antibodies (Fig. 6b and Supplemental Fig. 4c). The subpopulation of neurons labeled during the Recall Test #1 in the observer chamber was significantly reactivated when the observer is subjected recall in the observer chamber a second time (Fig. 6d), while we did not see the reactivation when we did not deliver shocks in the demonstrator during observational CFC (Fig. 6c). The subpopulation of neurons labeled during the Recall Test #1 in the demonstrator chamber is also reactivated when the observer is subjected to recall in the demonstrator chamber a second time (Fig. 6e). Crucially, the subpopulation of neurons that are activated by recall of observational CFC memory in the observer and demonstrator chambers do not overlap (Fig. 6f). To directly compare the reactivation levels between the 4 experimental groups, we calculated the fold change by examining the actual percentages and chance levels of Arc⁺mKate2⁺ PLC neurons, as we previously demonstrated[9] and found that the fold change in the Obs/Dem group was significantly lower than the Obs/Obs and Dem/Dem groups (Fig. 6g). These suggest that there are distinct subpopulations of mPFC neurons that are associated with recall of observational CFC memory in either the observer or demonstrator chamber at the remote time point.

## Discussion

In this study, we found that observational CFC memory is long-lastingly maintained, and can be recalled in the observer chamber at both the recent and remote time points (Fig. 1). On the other hand, observational CFC memory can be recalled in the demonstrator chamber only at remote time point (Fig. 1). dHPC activity is necessary

for recall of observational CFC memory in the observer chamber at the recent time point (Fig. 2), whereas the mPFC-BLA pathway is necessary for recall of observational CFC memory in the observer chamber at the remote time point (Fig. 3). Surprisingly, recall of observational CFC memory in the demonstrator at the remote time point also requires the mPFC-BLA pathway (Fig. 4), and the subpopulation of mPFC neurons activated during initial observational CFC is reactivated by recall of observational CFC memory in both the observer and demonstrator chambers at the remote time point. (Fig. 5). Finally, we demonstrated that there are different engram cell subpopulations within the mPFC that are associated with recall of observational CFC memory in the observer or demonstrator chamber at the remote time point even though these two subpopulations emerged from the neurons activated during observational CFC (Figs. 6 and 7).

Our model introduces four concepts. First, we demonstrated that observational CFC memory is long-lasting, which allowed us to explore the neural circuit mechanisms for remote memory recall of observational CFC memory. We then identified that observational CFC memory and direct CFC memory share the same neural circuits. In recall of observational CFC memory at the recent time point, similar to recall of direct CFC memory, neural activity of dHPC, specifically dDG and dCA1 subregions, is elevated and necessary (Fig. 2d, g, k). Neural activity of mPFC, on the other hand, is elevated and necessary for recall of observational CFC memory at the remote time point (Fig. 3), similar to remote recall of direct CFC memory. We specifically examined neural activity in mPFC by considering activation of mPFC subregions such as the SOM, ACAd, PLC, and ILC in both the observer (Fig. 3a–f) and demonstrator (Fig. 4d–g) chamber. Recall of observational CFC memory at the remote time point in the observer chamber broadly activated the SOM, ACAd, and PLC (Fig. 3c–f), whereas recall of observational CFC memory at the remote time in the demonstrator chamber only activated the PLC (Fig. 4d–g). Because there are robust local connections within the mPFC[79–81], the SOM is involved in perceptual behavior[79,82], and the ACAd and PLC are involved in recall of direct CFC memory[21,27,83], we speculate that recall of observational CFC memory in the observer chamber at the remote time point may be triggered by activation of a local mPFC circuit. In contrast, because

**a**

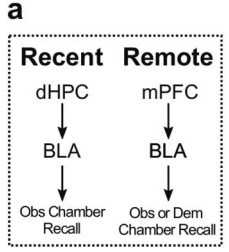

**b**

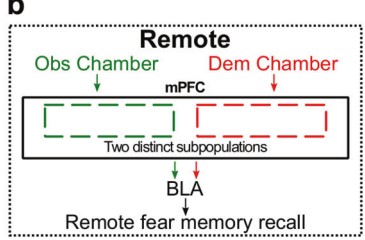

**Fig. 7 | Summary of Observational CFC neural mechanisms. a** Recent and remote recall of observational CFC memory in observer (Obs) or demonstrator (Dem) chamber. **b** Distinct neural populations in mPFC that regulate recall of observational CFC memory at the remote time point in the observer or demonstrator chamber.

only the PLC is activated by recall of observational CFC memory in the demonstrator chamber at the remote timepoint, we speculate that upstream inputs to the PLC such as thalamus, entorhinal cortex, or a combination of different regions, which are activated during recall of direct CFC memory at the remote time point[21,52], may trigger PLC neurons to elicit recall of observational CFC memory in the demonstrator chamber. Consistent with previous reports that demonstrate that the ILC is involved with extinction learning of fearful associations[84–86], we found that the ILC was not activated in by recall of observational CFC memory in the observer or demonstrator chamber at either time point (Figs. 3f and 4g). Although we did not directly investigate the role of more posterior subregions of the ACC, which are anatomically and functionally distinct than the region of the mPFC that we examined[87,88], previous reports found that ACC is dispensable for recall of observational CFC memory in the observer chamber at the recent time point[5] and is necessary for remote recall of direct CFC memory at the remote time point[27,89]. Therefore, we speculate that the ACC may broadly contribute to recall of observational CFC memory at the remote time point.

What are the specific neural mechanisms within the dHPC and mPFC that encode observational CFC Memory? Fear memory engram cells in dHPC and mPFC regulate the recent and remote recall of direct CFC memory, respectively[21,46,52,53,63,69,71–73]. Given the similarities between the role of dHPC in recent recall of observational CFC memory and direct CFC memory, we speculate that increased Arc positivity in dHPC during recent recall of observational CFC memory (Fig. 2d, g) indicates the reactivation of observational fear memory engram cells formed during observational CFC. In mPFC, similar to direct CFC memory[21,53], we found that an engram cell subpopulation in PLC is generated during observational CFC, which is then reactivated during recall of observational CFC memory in the observer and demonstrator chambers (Fig. 5n, p). These data suggest that fear memory engram cells associated with a context can be generated vicariously through the other's aversive experience.

Systems consolidation of a memory is the time-dependent process by which the neural circuits encoding a memory reorganize over time, resulting in transformation from detailed episodic memory to schema and gist-like semantic memory[31,32,34–36,63,67,90–92]. This type of memory transformation has been observed in many animal species including rodents, non-human primates, and humans[34–45,93]. In rodents, while transformation of direct CFC memory has been characterized[37–47], transformation of observational CFC memory has not been explored. Therefore, we considered whether observational CFC memory is transformed over time or, alternatively, if observational CFC memory remains constant by examining recall of observational CFC memory at the recent or remote time points in the demonstrator chamber. Unlike in the observer chamber, there was no head direction preference during freezing bouts in the recall test (Supplemental Fig. 1c, d) and observer freezing levels in the

demonstrator chamber did not correlate with freezing levels during observational CFC (Supplemental Fig. 2a, b), which suggests a difference in the quality of the recall of observational CFC memory in the observer or demonstrator chambers. By using the place preference chamber, we found that observers subjected to observational CFC neither exhibited any fear response in the demonstrator chamber nor chamber preference compared with the non-shock group at the recent time point (Fig. 1o). At the remote time point, however, observers had an elevated fear response in the demonstrator chamber and avoided the demonstrator chamber compared with the non-shock group (Fig. 1p). Moreover, there was no difference in freezing levels between observers subjected to observational CFC and observers in the non-shock group in a novel context (Fig. 1k) or a different context that was similar to the demonstrator chamber (Fig. 1m). These results suggest that after the systems consolidation, observers acquire the knowledge that the demonstrator chamber is dangerous. To inhibit the systems consolidation of observational CFC memory, we blocked the early tagging by inhibiting mPFC activity during observational CFC and found that the inhibition caused impairment of remote memory formation of observational CFC (Fig. 5j, k). These data show that systems consolidation induces a flexible recall strategy for observational CFC memory and enables animals to make new inferences about previously encountered situations, demonstrating the advantage of systems consolidation of memory in an animal model.

Recall of observational CFC memory in the observer and demonstrator chambers are fundamentally different behavioral processes. Therefore, we predicted that the underlying neural circuit mechanisms that regulate recall of observational CFC memory in the observer or demonstrator chambers would be different. Since dHPC neurons regulate egocentric and allocentric spatial representation depending on the situation[42,46,94,95], we initially predicted that dHPC activity may be required for recall of the observational CFC memory in the observer or demonstrator chamber even at remote time points. However, we found that dDG and dCA1 neural activity were not activated during recall of observational CFC memory in the observer and demonstrator chamber at remote time point (Figs. 2d, g and 4b, c), rather recall of observational CFC memory in both chambers at the remote time point is regulated by the mPFC-BLA pathway (Figs. 3g–i and 4h, i). We also found that the activation of subpopulation of mPFC neurons during observational CFC is reactivated during recall of the remote observational CFC memory in both the observer and demonstrator chamber (Fig. 5l–p). These lines of evidence strongly suggest that the neural circuit mechanisms that regulate recall of observational CFC memory in the observer or demonstrator chambers are quite similar. A remaining question then was how do the neural circuits that regulate recall of observational CFC memory diverge to produce these different cognitive processes? mPFC neural ensembles that encode a fear experience can change over time during memory consolidation[72], so one possibility is that different observational CFC engram cell populations within the mPFC may emerge during memory consolidation and thus differentially regulate recall of observational CFC memory in the observer and demonstrator chamber. To examine this, we labeled the subpopulation of mPFC neurons that were activated during recall of observational CFC memory in the observer or demonstrator chamber, subjected observers to recall of observational CFC memory a second time in the observer or demonstrator chamber, and examined the overlap between the activation of these subpopulations of neurons in PLC (Fig. 6a, b). We found that that there were multiple subpopulations of neurons that were differentially activated by recall of observational CFC memory in the observer or demonstrator chamber at the remote time point (Fig. 6f), suggesting that, during consolidation of observational CFC memory, an additional subpopulation emerged from the original population of mPFC neurons that were activated during observational CFC. We speculate that observers, after systems consolidation of observational CFC memory,

obtain a wider-scaled representation of the observational CFC context that encompasses both the observer and demonstrator chambers, which is represented by the two subpopulations of mPFC neurons that are associated with the observer or demonstrator chamber (Fig. 7). Furthermore, the emergence of the mPFC subpopulation associated with the demonstrator chamber during systems consolidation may suggest a neural mechanisms for extraction of the knowledge that the demonstrator chamber is dangerous from the observational CFC episode, similar to what has been reported in humans[96,97], which would thus enable observers to develop fear to the demonstrator chamber at the remote time point. Determining neural circuits and cell populations that distinctly regulate flexible recall strategies of observational CFC memory significantly advances our understanding of the ecological benefits in animals for memory consolidation.

## Methods

### Mice
C57BL/6J or TRE-Cre transgenic male mice between 9 weeks and 20 weeks of age were used for all experiments. Mice were group housed with littermates (2–5 mice per cage) for a minimum of 1 week prior to experiments in a 12 h (6 a.m.–6 p.m.) light/dark cycle, with food and water available ad libitum. All experiments were conducted during the light cycle. For virus-mediated activity-dependent cell labeling experiments, mice were maintained on a doxycycline (on-Dox) diet (40 mg/kg, Bio-serv) at least 1 week prior to stereotaxic surgery and were continuously fed this diet except for specified off-Dox days. All procedures relating to experimental treatments and mouse care conformed to NIH and Institutional guidelines, and were conducted with the approval of the UT Southwestern Institutional Animal Care and Use Committee (IACUC).

### Observational contextual fear conditioning (observational CFC)
A contextual fear conditioning apparatus (Med Associates) was modified to create two chambers (observer chamber and demonstrator chamber, both 15 cm W × 20 cm D × 20 cm H) divided by a non-perforated transparent plexiglass partition[9]. In the observer chamber, there was an opaque plexiglass floor, whereas in the demonstrator chamber there was an exposed stainless-steel rod floor. An observer and an unfamiliar demonstrator, which the observer was never exposed to prior to the start of the experiment, were put into their respective chambers. Observers and demonstrators were allowed to explore their chambers during the 5 min habituation period. Subsequently, the demonstrator was subjected to a 1.0 mA, 2-s foot shock with a 10 s shock interval for a total of 24 shock trials during the 4 min shock period. Non-shock control observers were subjected to the same procedures as observers subjected to observational CFC, except the demonstrator did not receive any foot shocks. Prior to observational CFC, observers regardless of group were separated into empty holding cages with bedding for 5–40 min before testing. Upon the completion of observational CFC, to minimize social transfer of fear, observers regardless of group were returned to their holding cages for 30–45 min.

### Recall test for observational CFC memory
After observational CFC or control conditions, observers were subjected to a recall test in different behavioral conditions. For all behavioral conditions (except the novel context and chamber preference conditions), the contextual fear conditioning apparatus was modified in the same way as in observational CFC. Observers were separated into holding cages 5–40 min before testing. A video tracking system (Med Associates) was used to record all behavior testing.

For recent and remote recall of observational CFC memory, 1 day (recent) or 28 days (remote) after observational CFC, observers were reintroduced into the observer chamber with a transparent partition, the demonstrator chamber with a transparent partition, or the observer chamber with an opaque (non-perforated) partition. For the novel context experiments, 28 days after observational CFC, observers were inserted into a completely novel context (dim white light, black plexiglass triangle insert with an opaque, white plastic floor, scented with 1% acetic acid, 23 cm W × 20 cm D × 22 cm H) or a novel context that was the same as the demonstrator chamber except the shock grid was covered by an opaque, white plastic floor. Duration of observer freezing response (in s) for 4 min was scored and compared between groups in each experimental condition. Freezing response was defined as a complete absence of movement, except for respiration. For the chamber preference test, 1 day (recent recall) or 28 days (remote recall) after observational CFC, observers were first put into a modified demonstrator chamber for 12 min, which was the same as the observational CFC demonstrator chamber except with a 7 cm W × 6 cm H square hole cut in the middle of the partition. Duration of time the observer spent in the demonstrator or observer chamber (s) and duration of observer freezing response in the observer chamber, demonstrator chamber, or during observational CFC (as a percentage, calculated with by the following formula: (time freezing in observer or demonstrator chamber (s)/total time spent in observer or demonstrator chamber (s)) × 100%) was scored and compared between groups in each experimental condition. Observer head direction was calculated as the percentage of time the observer's head direction was oriented in a particular direction during freezing. All behavior videos were analyzed using the Behavioral Observation Research Interaction Software (BORIS)[98].

### Adeno-associated viral vectors and drugs
$AAV_{2/8}$-CaMKIIα:hM4Di-mCherry, with a titer of $2.4 \times 10^{13}$ genome copy/mL, was acquired from Addgene (a gift from Bryan Roth, #50477). $AAV_{2/5}$-CaMKIIα:mCherry, with a titer of $2.9 \times 10^{12}$ genome copy/mL, was acquired from the UNC Vector Core (#AV4809D). $AAV_{2/5}$-CaMKIIα:eArchT3.0-eYFP, with a titer of $4.0 \times 10^{12}$ genome copy/mL, was acquired from the UNC Vector Core (#AV4883). $AAV_{2/5}$-CaMKIIα:eYFP, with a titer of $7.1 \times 10^{12}$ genome copy/mL, was acquired from the UNC Vector Core (#AV4808c). AAV-PRAM-d2tTA:TRE-NLS-mKate2 was acquired from Addgene and was serotyped with $AAV_9$ coat proteins and packaged at the University of Texas Southwestern Medical Center to make $AAV_{2/9}$-PRAM:d2tTA-TRE:NLS-mKate2 with a titer of $1.1 \times 10^{12}$ genome copy/mL (a gift from Yingxi Lin, #84474)[78]. $AAV_{2/8}$-hSyn-DIO-HA-hM4Di-IRES-mCitrine was obtained acquired from Addgene (a gift from Bryan Roth, 50455) with a titer of $3.1 \times 10^{13}$ genome copy/mL. Clozapine N-Oxide (CNO, Enzo Life Sciences), was dissolved in saline with a concentration of 4 mg/mL.

### Stereotaxic surgery
All surgeries were conducted using asceptic technique and conformed to NIH and UT Southwestern IACUC guidelines. A digital stereotax (David Kopf Instruments) with an attached stereomicroscope (Leica) was used for all surgeries. Anesthesia in the mice was induced with 4% isoflurane, and mice were maintained with 1–2.5% isoflurane for the duration of the stereotaxic surgery. The fur on top of the mouse's skull was shaved, and the scalp was sterilized with Povidone-Iodine and 70% Isopropyl Alcohol (Dynarex). A vertical incision in the scalp was made, and a drill burr (0.5 mm diameter, Fine Scientific Tools) with a micro-drill (Harvard Apparatus) was used to make a small hole directly above injections sites. A 10 μL Hamilton microsyringe filled with mineral oil and with a glass micropipette (Drummon Scientific) filled with mineral oil was used to complete all microinjections. Microinjection speed and volume was controlled by a microsyringe pump (World Precision Instruments). The micropipette was slowly lowered to the target site and a volume ranging from 300 nL to 500 nL was injected depending on the experiment. All microinjections were administered at a speed of 3.0 nL/s. Upon the completion of a microinjection, the micropipette remained in the target site for 5 min and then was slowly retracted.

After the injection, we sutured the skin to close incision location. After the completion of surgery, mice were administered meloxicam (2 mg/kg) as an analgesic, and remained on the heating pad until fully recovered from the anesthesia. Mice recovered for a minimum of 1 week before returning to group housing with cagemates. Post-mortem histology was performed to verify target sites.

## Histology and immunohistochemistry (IHC)

Mice were deeply anesthetized with a cocktail of ketamine (75 mg/kg)/dexmedetomidine (1 mg/kg) cocktail and then transcardially perfused with 4% paraformaldehyde (PFA) in PBS. Brains were extracted and post-fixed overnight in 4% PFA in PBS at 4 °C and then sectioned at a thickness of 50 μm using a vibratome (Leica). For IHC, sections were incubated in PBS solution containing 0.4% Triton-X and 10% normal goat serum (PBS-T) for 1 hour. Primary antibodies were then added to the PBS-T solution and sections were incubated overnight at 4 °C. Primary antibodies used were rabbit anti-Arc antibody (1/500, Synaptic Systems, 156003) and mouse anti-NeuN antibody (1/1000, Millipore Sigma, MAB377). The sections were then rinsed with PBS for 3 × 10 min, and subsequently incubated for 2 hours with AlexaFluor 488 goat anti-rabbit IgG (A11008), AlexFluor 546 goat anti-rabbit IgG (A11010), or AlexaFluor 633 goat anti-mouse IgG (A21050) conjugated secondary antibodies (1/500, ThermoFisher Scientific) in PBS-T. Sections in experiments in Figs. 5l–p and 6 were incubated for 3 h after washing with NeuroTrace 435/455 Blue Fluorescent Nissl Stain (1/500, ThermoFisher Scientific). The sections were then rinsed in PBS for 3 × 10 min, and mounted onto glass slides in VECTASHIELD medium (Vector Laboratories). A subset of sections were counterstained using DAPI (1/1000, ThermoFisher Scientific). For dCA1, fluorescent images were obtained using a Zeiss LSM800 confocal microscope with Airyscan using the ×25 objective. For all brain regions, fluorescent images were obtained using a Zeiss AxioImager M2 microscope with Apotome using the ×2.5, ×5, and ×10 objectives. All images were processed using the Zen Blue software.

## Quantification of Arc expression in dorsal dentate gyrus (dDG), dorsal CA1 (dCA1), basolateral amygdala (BLA), and medial prefrontal cortex (mPFC) subregions

NeuN positive and Arc (activity-regulated cytoskeleton-associated protein) positive cells were quantified in each brain region using the Cell Counting plugin in ImageJ. The percentage of Arc$^+$ neurons in each brain region was calculated out of total neurons (Arc$^+$NeuN$^+$/Total NeuN$^+$ × 100%) and compared between groups. 2-6 coronal sections for dDG (AP: −2.00), dCA1 (AP: −2.00), BLA (AP: −1.40), and mPFC (AP: +2.10 to +1.50 containing Secondary Motor Area (SOM), Anterior Cingulate Area dorsal part (ACAd), Prelimbic Cortex (PLC), and Infralimbic Cortex (ILC), as examined in a previous report and indicated in the Allen Brain Institute Atlas[21,81]) were collected per mouse. In Fig. 2d, there was a total of 75,396 (Recent Non-Shock: 17,365; Recent Observational CFC: 21,617; Remote Non-Shock: 16,969; Remote Observational CFC: 19,445) NeuN$^+$ cells in dDG. In Fig. 2g, there was a total of 10,173 (Recent Non-Shock: 2842; Recent Observational CFC: 2436; Remote Non-Shock: 2393; Remote Observational CFC: 2502) NeuN$^+$ cells in dCA1. In Fig. 3c, there was a total of 30,754 (Recent Non-Shock: 9543; Recent Observational CFC: 10,131; Remote Non-Shock: 6395; Remote Observational CFC: 4685) NeuN$^+$ cells in SOM. In Fig. 3d, there was a total of 18,141 (Recent Non-Shock: 5739; Recent Observational CFC: 5854; Remote Non-Shock: 3630; Remote Observational CFC: 2918) NeuN$^+$ cells in ACAd. In Fig. 3e, there was a total of 28,804 (Recent Non-Shock: 8860; Recent Observational CFC: 9249; Remote Non-Shock: 6446; Remote Observational CFC: 4249) NeuN$^+$ cells in PLC. In Fig. 3f, there was a total of 21,786 (Recent Non-Shock: 7528; Recent Observational CFC: 6700; Remote Non-Shock: 5112; Remote Observational CFC: 2446) NeuN$^+$ cells in ILC. In Fig. 3j, there was a total of 25,440 (Light Off/eYFP: 3092; Light Off/eArchT: 5053; Light On/

eYFP: 9353; Light On/eArchT: 7942) NeuN$^+$ cells in BLA. In Fig. 4b, there was a total of 73,738 (Recent Non-Shock: 13,600; Recent Observational CFC: 15,968; Remote Non-Shock: 23,323; Remote Observational CFC: 20,847) NeuN$^+$ cells in dDG. In Fig. 4c, there was a total of 10,860 (Recent Non-Shock: 2635; Recent Observational CFC: 2226; Remote Non-Shock: 3120; Remote Observational CFC: 2879) NeuN$^+$ cells in dCA1. In Fig. 4d, there was a total of 32,479 (Recent Non-Shock: 8565; Recent Observational CFC: 7442; Remote Non-Shock: 9348; Remote Observational CFC: 7124) NeuN$^+$ cells in SOM. In Fig. 4e, there was a total of 17,675 (Recent Non-Shock: 5124; Recent Observational CFC: 3986; Remote Non-Shock: 4743; Remote Observational CFC: 3822) NeuN$^+$ cells in ACAd. In Fig. 4f, there was a total of 30,699 (Recent Non-Shock: 8358; Recent Observational CFC: 6480; Remote Non-Shock: 9278; Remote Observational CFC: 6583) NeuN$^+$ cells in PLC. In Fig. 4g, there was a total of 26,873 (Recent Non-Shock: 7762; Recent Observational CFC: 5722; Remote Non-Shock: 8229; Remote Observational CFC: 5160) NeuN$^+$ cells in ILC. In Fig. 5b, there was a total of 16,766 (Non-Shock: 9234; Observational CFC: 7532) NeuN$^+$ cells in SOM. In Fig. 5c, there was a total of 7515 (Non-Shock: 3879; Observational CFC: 3636) NeuN$^+$ cells in ACAd. In Fig. 5d, there was a total of 15,288 (Non-Shock: 7710; Observational CFC: 7578) NeuN$^+$ cells in PLC. In Fig. 5e, there was a total of 11,561 (Non-Shock: 5898; Observational CFC: 5663) NeuN$^+$ cells in ILC.

## Chemogenetic inhibition

Designer receptors exclusively activated by designer drugs (inhibitory DREADDs) were used to inhibit dorsal hippocampus (dHPC) and medial prefrontal corte (mPFC). dHPC was bilaterally injected with 500 nL per side of AAV$_{2/8}$-CaMKIIα:hM4Di-mCherry (experimental) or AAV$_{2/5}$-CaMKIIα:mCherry (control) aimed at the following coordinates relative to bregma: AP: −2.00 mm; ML: ±1.50 mm; DV: −1.40 mm. mPFC was bilaterally injected with 500 nL of AAV$_{2/8}$-CaMKIIα:hM4Di-mCherry (experimental) or AAV$_{2/5}$-CaMKIIα:mCherry (control) aimed at the following coordinates relative to bregma: AP: +2.00 mm; ML: ±0.30 mm; DV: −2.00 mm. Observers recovered for 2 weeks before being group housed with cagemates. 30 minutes prior to behavior testing in experiments in which dHPC or mPFC was inhibited, observers were injected with CNO (4 mg/kg dose). Our previous study demonstrated that the CNO treatment significantly reduced the neural activity in hippocampal DREADD-expressing neurons in vitro and in vivo[9].

## Optogenetic inhibition of mPFC terminals in BLA

mPFC was bilaterally injected with 400 nL per side of AAV$_{2/5}$-CaMKIIα:eArchT-eYFP (experimental) or AAV$_{2/5}$-CaMKIIα:eYFP (control) aimed at the following coordinates relative to bregma: AP: +2.00 mm; ML: ±0.30 mm; DV: −2.00 mm. Optical fibers (Doric Lenses) targeting BLA were bilaterally implanted at the following coordinates relative to bregma: AP: −1.40 mm; ML: ±3.40 mm; DV: −5.00 mm. Two screws and dental cement (C&B Metabond, Parkell) were used to secure the optical fibers. Optical fibers were further protected using the top part of an Eppendorf tube that was attached and secured with dental cement. Observers recovered for 1 week before being group housed with cagemates. Observers were then subjected to observational CFC. 28 days later, observers were subjected to remote recall of observational CFC memory in the observer chamber with optogenetic inhibition. Observers were bilaterally connected to a 532 nm laser (UltraLasers), which was controlled by a function generator (Siglent Technologies). Green light stimulation (15 mW, each hemisphere) was delivered to the observer during the behavior test with the following stimulation protocol (which was used to prevent overheating/overstimulation): 13 s of laser stimulation on and 2 s of laser stimulation off. We also included light-off control groups in which eYFP and eArchT observers were subjected to recall of observational CFC memory without light stimulation. We quantified NeuN positive and Arc

positive neurons in BLA in some eYFP and eArchT mice (see "Quantification of Arc expression" above). In a subset of mice, one day later, both eYFP and eArchT observers were tested for remote recall of observational CFC memory in the demonstrator chamber using the same optogenetic stimulation protocol.

## Engram cell labeling in mPFC

For long-term labeling of mPFC engram cells in Fig. 5l–p, TRE-Cre mice bilaterally injected with a 400 nL cocktail per side containing 200 nL of AAV$_{2/9}$-PRAM:d2tTA-TRE:NLS-mKate2 and 200 nL of AAV$_{2/8}$-hSyn-DIO-HA-hM4Di-IRES-mCitrine aimed at the following coordinates relative to bregma: AP: +2.00 mm; ML: ±0.30 mm; DV: −2.00 mm. 3 weeks after injection, observers were subjected to observational CFC in the off-Dox condition to label mPFC engram cells, and were then immediately put back on the Dox diet. Observers were then subjected to recall of observational CFC memory in the observer or demonstrator chamber at the remote time point, and were perfused 1 h after testing. Neurons that were activated by observational CFC were labeled with mCitrine and neurons that were activated during the recall test were labeled with Arc. We did not detect mKate2+ neurons at the remote time point because of degradation of the mKate2 signal from the on-Dox conditioning. 2-6 coronal sections for mPFC (AP: +2.10 to +1.50) were collected per mouse, of which the PLC subregion was quantified. We calculated the percentage of Arc$^+$mCitrine$^-$ and Arc$^+$mCitrine$^+$ in each mouse. Fold change analysis was calculated using the following formula: ((Average percent overlap of Arc$^+$mCitrine$^+$ cells)/(Chance percent overlap of Arc$^+$mCitrine$^+$ cells) * 100%). In Fig. 5n, there was a total of 11,602 Nissl$^+$ cells in PLC. In Fig. 5p, there was a total of 12,120 Nissl$^+$ cells in PLC.

For mPFC engram cell reactivation during recall of observational CFC memory in Fig. 6, wildtype mice were subjected to observational CFC (or non-shock control condition). One week later, mice were bilaterally injected with 300 nL of AAV$_{2/9}$-PRAM:d2tTA-TRE:NLS-mKate2 aimed at the following coordinates relative to bregma: AP: +2.00 mm; ML: ±0.30 mm; DV: −2.00 mm. 3 weeks later, in the off-Dox condition, mice were subjected recall of observational CFC memory in the observer or demonstrator chamber. Immediately after testing, mice were put back on-Dox. 1 day later, mice were again subjected to recall of observational CFC memory in the observer or demonstrator chamber, and were perfused 1 hour after testing. We adopted the following shorthand to describe each testing group. Obs #1/Obs #1, Non-Shock (observers in the non-shock condition that are subjected to recall of observational CFC memory in the observer chamber in the off-Dox condition, and recall again in the observer chamber in the on-Dox condition the next day). Obs #1/Obs #1, Obs CFC (same as Obs #1/Obs #1, except these observers were subjected to observational CFC). Dem #1/Dem #2, Obs CFC (observers in the observational CFC condition that are subjected to recall of observational CFC memory in the demonstrator chamber in the off-Dox condition, and recall again in the demonstrator chamber in the on-Dox condition on the next day). Obs #1/Dem #2, Obs CFC (observers in the observational CFC condition that are subjected to recall of observational CFC memory in the observer chamber in the off-Dox condition, and recall again in the demonstrator chamber in the on-Dox condition on the next day). Neurons that were activated during the first recall test were labeled with mKate2, and neurons that were activated during the second recall test were labeled with Arc. Two to six coronal sections for mPFC (AP: +2.10 to +1.50) were collected per mouse, of which the PLC subregion was quantified. We calculated the percentage of Arc$^+$mKate2$^-$ and Arc$^+$mKate2$^+$ in each mouse. Fold change analysis was calculated using the following formula: ((Average percent overlap of Arc$^+$mKate2$^+$ cells)/(Chance percent overlap of Arc$^+$mKate2$^+$ cells) × 100%). In Fig. 6c, there was a total

of 7363 Nissl$^+$ cells in PLC. In Fig. 6d, there was a total of 11,993 Nissl$^+$ cells in PLC. In Fig. 6e, there was a total of 7053 Nissl$^+$ cells in PLC. In Fig. 6f, there was a total of 11,225 Nissl$^+$ cells in PLC.

## Quantification and statistical analysis

Data are presented as box plots with the minimum, first quartile, median, third quartile, and maximum values indicated, correlations, or line graphs with standard error of the mean. Statistical methods were not used to predetermine sample sizes in experiments; sample sizes were selected based on what is conventional for the field, which previous studies determined were sufficiently powerful to detect meaningful differences (or lack of differences) between groups[5,9,21,99]. Mice were randomly assigned to groups, and experimenters were blinded to the conditions of experiments prior to completing data analysis. Experiments were conducted twice to verify that similar findings were obtained, and results were then pooled. Correlation, One-way ANOVA with Tukey test, 2 × 2 between-subjects ANOVA with Bonferroni or Tukey test, 2 × 2 mixed ANOVA with Bonferonni test, unpaired $t$ test (two-tailed or one-tailed), and paired $t$ test (two-tailed or one-tailed) were used when appropriate. If the variances between groups in an unpaired $t$ test were significantly different, we performed the Mann–Whitney $U$-test (two-tailed or one-tailed) instead. Outliers were detected using the Grubbs' method with the threshold for removal set to Alpha = 0.01. Graphpad Prism 9 software was used to calculate $P$ values for frequentist statistical analyses. The null hypothesis was rejected at the $P < 0.05$ level. JASP version 0.17.1 with default priors was used to perform Bayesian analyses for all statistical tests[100]. Bayesian analyses were used to determine if an observed lack of an effect was due to evidence of absence or if a lack of an effect was due to insufficient evidence[101]. All statistics (including Bayesian factors) and $N$s are presented in Supplemental Table 1.

## Reporting summary

Further information on research design is available in the Nature Portfolio Reporting Summary linked to this article.

## Data availability

Requests for materials and correspondence should be made to the lead author, Takashi Kitamura (takashi.kitamura@utsouthwestern.edu). The datasets generated during and/or analyzed during the current study are available from the corresponding author upon request. Source data are provided with this paper.

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

## Acknowledgements

We thank all members and collaborators of the Kitamura Laboratory for their support. We also thank Dr. Stephen Maren about interpretation for data and Dr. Shari Birnbaum for advice on behavioral approaches. This work was supported by Endowed Scholar Program to T.K., Faculty Science and Technology Acquisition and Retention Program to T.K., and National Institute of Mental Health to T.K. (R01MH125916) and J.I.T. (F32MH119721).

## Author contributions

J.I.T., S.K.O., and T.K. contributed to the study design. J.I.T. and J.Y. conducted the experiments in the manuscript. J.I.T. performed the data analyses. J.I.T., H.O., S.K.O., and T.K. contributed to the interpretation for data analyses. J.I.T. and T.K. wrote the paper. All authors approved the final manuscript.

## Competing interests

The authors declare no competing interests.
