## [Peer Review File · Nature Communications]

Systems consolidation induces multiple memory engrams for a flexible recall strategy in observational fear memory in male miceEditorial Note: Parts of this Peer Review File have been redacted as indicated to remove third-party material where no permission to publish could be obtained.

REVIEWER COMMENTS

Reviewer #1 (Remarks to the Author):

In their manuscript, Terranova et al. explore what mice learn from witnessing other mice receive footshocks, and how this evolves over time after observation. In particular, two mice are placed in opposite compartments of a shuttle box, and the observer mouse on one side sees the demonstrator receive footshocks in the other. Thereafter, the observer mice are placed back into the observer compartment, the demonstrator compartment or a new context, either on day 1 after observation or on day 28. Behaviorally, the core finding is that mice freeze in the observer compartment all the time (day 1 & day 28), when placed in the demonstrator compartment, they freeze on day 28 but not day 1, when placed in a different compartment altogether (either a differently shaped box or the shuttle box with an opaque wall), they do not freeze at all. In agreement with the literature on memory, this suggests a widening of fear over time, with a representation very specific to where the mouse was when shocked on day 1 (observer compartment only), that becomes a bit wider (observer + similar demonstrator compartment but not the more different control box or opaque divider box) by day 28.

Neuroscientifically, arc positive cell counts suggest that the dorsal hippocampus is recruited when animals are replaced in the observer compartment 1d after shock observation, but not when replaced 28d later, with chemogenetic inhibition of the dorsal hippocampus interfering with freezing in observer compartment on day 1 but not day 28. In contrast, for the medial prefrontal cortex, deactivation using NMDA lesions, reduces freezing on day 28 but not day 1, and this is true also when deactivating the connections from mPFC to the BLA.

In contrast, when placed in the demonstrator compartment, arc was not increased in the dorsal hippocampus or mPFC on day 1, but was on day 28, significantly in the mPFC, and at trend level in the the dorsal hippocampus. mPFC->BLA inhibition triggered reduced freezing in the demo compartment on day 28.

While previous data had already shown that freezing on day 1 is stronger in the observation compartment than in a different context and other studies have shown the importance of mPFC, hippocampus and amygdala in this phenomenon, the current study provides interesting additions to the existing literature that are in line with the memory literature on contextual fear conditioning: that the specificity of the memory is reduced over time, and that that goes hand in hand with a reduced importance of the hippocampus. In addition, showing that the animals' preference for the two compartments changes over time is interesting. However, there are several issues that would need to be addressed.

a) allocentric vs egocentric. The data presented in the paper shows that freezing that is specific to the exact context in which shocks were observed on day 1 generalizes to the similar demonstrator compartment context, but not to the very different control context or a box with an opaque divider. This 'gist'-like generalization to similar contexts over time is in line with the memory literature more generally. The authors however interpret it as reflecting a transition from egocentric to allocentric coding. These terms have very strong meanings in the visual neuroscience literature, and the authors do not sufficiently support the notion of an allocentric representation above and beyond the simpler concept of a gradual generalization to other, similar contexts. Framing the manuscript around these notions, seems misleading, as it suggests that there is proof of an allocentric representation suggesting a viewpoint independent but specific representation of space, that the data simply does not seem to support. We therefore recommend to remove these terms throughout, and replace them with more parsimonious terms: effects restricted to the observer compartment vs effects generalizing to the demonstrator compartment. Perhaps in the discussion, the authors can mention that this data could be compatible with a generalization gradient, in which the demonstrator compartment is more similar to the observer compartment than the other context or a box with opaque divider, or, alternatively, with the notion that the mPFC contains a representation of the entirety of the box,

including both compartments, while the dHPC has a more fine grained representation restricted to the observer compartment, and that the importance of these two representational granularities switches over time, with behavior dominated by the finer-grained spatial representation in the dHPC on day1, and by the wider-scaled representation of the entire box on day 28. Still, a less fine grained representation is far from being 'allocentric'. Statements like "Egocentric and allocentric recall of observational CFC memory are fundamentally different 17 behavioral processes because, in egocentric recall, the observer recalls observational CFC memory 18 from their perspective, whereas, in allocentric recall, the observer recalls observational CFC 19 memory from the perspective of the demonstrator's context." really seem to go beyond the data: how do we know that the memory at play at d28 is from the demonstrators perspective?

b) interactions: much of the statistics in the paper are restricted to t-tests, and the attribution of a particular function to a brain regions is based on the presence of an effect in one and an absence in another. This is statistically not the appropriate test for specificity: specificity corresponds to the demonstration of an interaction, i.e. that the effect is larger in one region or at one time point than another. The authors should thus replace the t-test with appropriate interaction tests (see Nieuwenhuis, S, B U Forstmann, and E.--J. Wagenmakers. "Erroneous Analyses of Interactions in Neuroscience: A Problem of Significance." *Nature Neuroscience* 14 (2011): 1105–7 for an explanation of this point). To illustrate some examples: Figure 1 (G-I) suggests that freezing is increased in the demonstrator compartment over time, by showing that the non-shock < Obs CFC at remote but not recent timepoints. What should be shown is timepoint x group interaction: is the group difference larger on d28 than d1. If this is not significant (it probably is, but if it isn't), it wouldn't be appropriate to suggest that the passing of time leads to an increase in freezing in the demo compartment. The same applies in Figure 2 for Arc in dorsal HPC, for the effect of chemogenetics on dHPC (larger in recent than remote rather than significant at recent but not remote), or Figure 3 and 4 when comparing remote and recent mPFC Arc expression, in the two compartments, Fig 5 when comparing the chemogenetic effects. In all these cases, conclusions can only be meaningfully drawn from these interaction effects. I understand that in some cases, the data may not be normally distributed, yet, ANOVAs are relatively permissive against modest deviations from their assumptions, which can be checked in a QQ plot. If data really is not normal, using a rank based method could be useful (as implemented in the CRAN package ARTool).

c) Evidence of absence: in several places, the manuscript interprets the lack of an effect (e.g. "There was no difference in Arc positivity between the observational CFC and the non-shock group in either dDG (Figure 4C, 4J) or dCA1 (Figure 4E, 4L) at the recent or remote time points"). To do so meaningfully, it is important to adjudicate whether the dependent variable provides evidence for H0 vs an effect too weak to be significant given a specific group size. This should be addressed by reporting Bayes factors in addition to the p values (see Keyesers, Christian, Valeria Gazzola, and Eric-Jan Wagenmakers. "Using Bayes Factor Hypothesis Testing in Neuroscience to Establish Evidence of Absence." *Nature Neuroscience* 23, no. 7 (July 29, 2020): 788–99. <https://doi.org/10.1038/s41593-020-0660-4>. For a tutorial on how to do this).

d) the group sizes are sometimes 10, sometimes 15, and it is unclear why such fluctuations exist. The methods section does not sufficiently explain these choices: "Statistical methods were not used to predetermine sample sizes in experiments; sample sizes were based on what is conventional for the field and previous studies". Why previous experiments would suggest N=9,10 or 15 based on different experiments is unclear.

Minor comments:

Figures: please add dots for each individual, and plot freezing always on the same scale for all experiments so that readers can judge the relative levels of freezing across the many experiments, and judge whether 'control' conditions in various cohorts really produced 'typical' levels of freezing. Otherwise, some control vs experimental differences could be driven by differences in the control

condition.

P5, l43: 'silencing of dHPC' should probably be mPFC.

Fig4, panel A, Mouse should probably be in the demo compartment also on day 1.

In the various figures, please replace absolute freezing time with %-freezing to facilitate comparison with other studies that may have monitored freezing over different durations.

Line 6, page 9 completion rather than competition likely

For Place preference test, animals always placed in demos chamber?

Fig5: In legend 'J' and 'H' should be interchanged.

P5l2 'prior to egocentric recall'. It would be good to provide more details about the timing of the lesion relative to shock observation and recall either in the text of figure 3.

P6: "at both the recent and remote time point". Add s at time points.

P7: "These results indicate that after the systems consolidation, observers acquire the ability to allocentrically recall observational CFC memory. ". This is an example of what needs to be reframed: generalization from the observer to the demonstrator compartment is a parsimonious explanation of the same data.

P8: "divided by a transparent plexiglass partition, as previously described ". Was the divider also perforated?

P11: "NMDA-Induced Excitotoxic Lesion mPFC was bilaterally injected with 150 nL per side of NMDA (25 mg/mL) or saline (control) aimed at the following coordinates relative to bregma: AP: +2.00 mm, ML: \pm 0.30 mm, DV: -2.00 29 mm. Observers recovered for 1 week before being group housed with cagemates". Still not quite clear to me, whether lesions were made prior to shock observation, or after observation but before recall? And how long before shock observation was the NMDA applied?

P11: "13 seconds of laser stimulation on and 2 seconds of laser stimulation off": was a rebound observed behaviorally during 2s off?

Fig1: A Mann-Whitney U is used for A-C, but a student t-test for some of the other tests. Panel J-K, No recall in novel environment is a critical test, and yet $P=0.07$. Here also the difference in N becomes intriguing, as the same effect size for $N=15$ would be significant.

Fig2: include a non-shock arc/NeuN micrograph for comparison. Also specify how arc% is calculated (double positive count (Arc&NeuN)/(NeuN)?

Fig3G: the Sal group clearly contained an outlier. Recalculate U value without the outlier.

Fig4: again include no-shock micrograph for comparison.

Fig4N: here the non-shock seems low compared to other panels, rather than obsCFC increased.

Fig5K: in the mcherry group, freezing should be similar to Fig 1l obs CFC. Is that the case? This also emphasizes the importance of having all freezing graphs on the same scale.

Reviewer #2 (Remarks to the Author):

Terranova et al. addressed the characteristics and neural mechanisms underlying observational CFC in one's own or a party's point of view, discovering that a certain amount of time must pass before allocentric recall becomes possible and that mPFC (and BLA) is involved in the recall. Although it is unclear now do the authors define mPFC (does it include not only PL/IL but also cingulate cortex or ACC?), this study introduces a new perspective on observational [fear] learning. Some findings are still primitive, but they are highly suggestive. My comments are as follows:

Major comments

1. About freezing in the Dem chamber under the Remote conditions (perhaps the most important finding herein). I would like to see not only the references of the upstream brain regions involved in egocentric and allocentric recalls, but also a discussion of why it takes so long to acquire the defensive behavior. Although this may be a general question in the systems consolidation, it was strange for me that it did not freeze in Dem Recent. If the authors assume that the shift of Dem Recent to Dem Remote is not a generalization, I am even more curious about the mechanism.

2. Regarding the consideration of CS-dependent (Result Page.3, L27-28). If it is CS-dependent, then elimination learning should be possible. What do the authors think is the case? Is there a difference in the freezing percentage between the early and late stages during the Recall test? More specifically, in general contextual fear conditioning, a temporal transition of increasing and decreasing freezing is observed. Did the authors see such a time-dependent phenomenon? Even if not, I would like to see a time change in the freezing percentage, rather than the total freezing time. This data would be an indicator of the type of information and timing to which mice exhibit freezing.

3. On the denial of the possibility of generalization (Result L39-44). I think that Fig.1J,K are important data. I was concerned about "scented with 1% acetic acid" (Methods Page 9, L 22). I think the author probably employed a common method, but I am wondering if the acetic acid itself might be a stimulus to increase activity (search behavior). I think that there is probably a floor effect, with $P = 0.07$, but I am more surprised that the Obs CFC group tends to have a lower freezing rate.

4. Fig.3 F; It is difficult to evaluate if only mPFC is lesioned. It is necessary to check the leak to see if the lesion affected other important areas such as ACC. I think that even in the representative image in Fig 5E, the injected area is too large to claim to be specific, and the authors cannot rule out that the data obtained in this study reflected indirect effects through other brain areas. The authors must convincingly prove the specificity of their neural manipulation. Since the roles of PL, IL, and ACC are very different, it is essential to distinguish these brain regions properly.

5. People have questioned whether ArchT can really suppress the activity of axon terminals. Although ArchT can suppress the generation of action potentials, in general, it does not have such a strong effect that it suppresses action potentials once generated. The authors have used this method in their previous reports too, but they should demonstrate that ArchT indeed works under their in vivo experimental conditions. Otherwise, this work will again be viewed with skepticism by researchers in the same field, as was the case in the previous reports.

6. In the Discussion (P7L28-30), the authors described " We also found that the activation of subpopulation of mPFC neurons during observational CFC is necessary for both egocentric and allocentric recall of the remote observational CFC memory (Figure 5H, 5K), suggesting that same population of mPFC neurons may regulate both egocentric and allocentric remote recall process". In Fig. 5, as in Fig. 3-4, by manipulating only the cells that project to BLA, the author can show that the "same" cell population is involved.

Minor comments

1. What is the reason for the difference in experimental conditions? 30-45min (Methods page 9; L 7-8) vs 5-40min (L14). Typo?
2. hm4Di is incorrectly described as AAV2/8 (Methods page 9; L 36-37) and mCherry as AAV2/5 (Page 11, L17-18)? The text and figure are correct.
3. The number of Arcs also seems to increase outside of mPFC ROI. Why? Please quantify them.
4. Fig. 4A; Is the position of the mouse a drawing error? Since it's Allocentric, shouldn't the mouse be on the grid?
5. Some graphical abstract would help the readers.
6. I would like to see some discussion on the difference between mPFC and ACC in terms of their contribution to observational CFC, since there is no mention of ACC in the Discussion or the Results although ACC is included in the Introduction (P1L43).
7. I do not understand why the author used NMDA-induced excitotoxicity instead of DREADD for the suppression of mPFC (Fig. 3). Is it because it's easy?

Reviewer #3 (Remarks to the Author):

The manuscript by Terranova et al. seeks to study the underlying neural mechanisms that contribute to empathic observational CFC memory. They provide data showing that observational CFC training can be consolidated into long-term memory and found different brain areas are responsible for either egocentric or allocentric retrieval of recent or remote observational CFC memory. The authors used a large variety of techniques to address their question, however, there are still a number of issues with the significance and approach of the paper.

1. During the observational CFC memory training session, what's the freezing level of the observer mice when demonstrator mice experience strong and lasting foot shocks? Is the freezing level or duration of gazing at the opposite chamber correlated with the recall performance afterward?
2. Head direction is pivotal to egocentric spatial encoding from the observer's perspective. What's the main head direction of observer mice behaving freeze during egocentric or allocentric recall of CFC?
3. In Fig2N, why the freezing time in remote recall (both mcherry and hm4d treated mice) are longer than the recent recall (Fig1 C and Fig 2M)?
4. Fig2 lacks Arc staining of no-shock group mice.
Fig3K lacks the BLA Arc level after laser stimulation in the no-shock group. Light-off control experiments are also needed.
5. Fig4A, placement of Obs mice in recall test seems wrong? It should be placed at the Dem chamber.
6. Page5 Line43 dHPC should be corrected by mPFC.
7. What impresses me most, in this study, is the neural circuit underlying observational CFC is almost the same with that of well-studied direct CFC memory recall. Maybe specific subpopulations of cells or projections from the same brain areas, or specific in-vivo firing patterns, contribute to the formation of observational CFC and to the perspective-dependent and time-dependent retrieval of observational CFC memory. It is a pity that the authors did not solve these problems any further.

Reviewer #4 (Remarks to the Author):

Memories for events initially depend on the hippocampus, but with time become increasingly dependent on extra-hippocampal regions for their expression. This time-dependent process of reorganization is known as systems consolidation, and has been especially characterized in rodent models using contextual fear conditioning tasks. The current manuscript by Terranova and colleagues examines whether systems consolidation occurs for another form of memory—observational fear memory—where rodents learn about environmental contingencies through observation of conspecifics. A particularly interesting aspect of this task is the potential to distinguish between ego- and allocentric recall (i.e., recall from the subject mouse's perspective vs. the demonstrator's mouse perspective). The work presented here shows that observational fear memories may be recalled egocentrically at recent and remote delays. However, they also make a case that observational fear memories may be recalled allocentrically at remote delays, suggesting there is some additional 'insight' that emerges with time and may track time-dependent changes in memory organization. With respect to these time-dependent changes, using IEG analyses and chemogenetic regional activation/silencing approaches they establish that recall of recent memories engages dorsal hippocampus, whereas remote recall engages mPFC-BLA projections. Additionally, they show that mPFC activity, at the time of encoding, is necessary for the formation of remote observational fear memories. This work is an important addition to the literature on systems consolidation, providing further support for a time-dependent shift from hippocampus to cortex in a novel learning paradigm. The experiments are carefully conducted, and the manuscript is well-written. That said, we identified two important issues that could be addressed in a revision.

1. Memory generalization vs. allocentric observational fear. In experiment presented in Fig. 1G-, the authors found that observer mice freeze in the demonstrator's chamber only at remote time-point, and this is interpreted as an example of allocentric observational fear. An alternative possibility, as acknowledged by the authors, is that this reflects memory generalization. In other words, the fear associated to the observer chamber can generalize to demonstrator chamber as the memory ages. To address this, authors exposed observational CFC mice to a novel context (Fig. 1J-K) and concluded absence of generalization due to low freezing at novel context. However, we have the following concern. Perhaps the novel context is very different, and generalization might be observed in a more similar context. Since the observer chamber and demonstrator chamber share so many features (especially that observer mice can see the demonstrator context) it might be the case that the demonstrator chamber is functioning as a similar context. Moreover, previous exposure to a context can accelerate generalization to it. The fact that observer mice are exposed to demonstrator context therefore could lead to a faster generalization of the fear memory for this context compared to a novel context. Finally, and most importantly, in experiment 1D, authors have shown that an opaque wall during the recall impedes the recall of the observational CFC. This outcome indicates that some features of the demonstrator chamber are being used for recall, so it would be much easier for mice to generalize the fear to this context rather than a novel context. One potential solution to distinguish allocentric recall from a generalized recall would be to add a third chamber. This way at the same time the experiment would include a no-shock demonstrator in chamber A, observer mouse in chamber B (at the center so the observer can see both of the demonstrators), and a shocked demonstrator in chamber C. At the time of recall the observer would be exposed to chamber A. If freezing level elevates this means the observer could not simply distinguish between the chambers and hence the memory was generalized. On the other hand, if the observer doesn't freeze in chamber A (while freezing to chamber C), then the conclusion is the mouse can distinguish the chambers and it only freezes in an allocentric situation. We appreciate that this type of experiment might not be possible, but we are at least interested in the authors' thoughts on this issue.

2. Global activation/inhibition vs. engram tagging. The authors chemogenetically activated/silenced entire regions. The application of engram cell specific manipulations would be interesting, especially with respect to the possibility to distinguish between ensembles encoding ego- vs. allocentric memory in the mPFC.

Minor

1- The freezing levels for some cases are reported in seconds. It would be better to report them all in

percentages.

2- Overall, the freezing levels are pretty low (around 10-20 seconds, or 4-8% in case of 4 min recalls). Authors should be cautious about how meaningful a 4% change in freezing could be for their behavior paradigm.

3- The rationale for using Mann-Whitney U-test for the first comparison of freezing levels (used only for fig 1.A-C) should be provided (t-tests were used for remaining figures)..

4- Line 43: I believe "dHPC" is a typing mistake, it should be "mPFC"

5- The authors have explored the essentiality of dHPC with inhibitory dreadds while they use NMDA lesioning for mPFC. What was the rationale for these choices and why not use the same method for both experiments?

Dear Editors and Reviewers,

We thank you very much for your time and effort in the review of our manuscript, “Systems consolidation induces multiple memory engrams for a flexible recall strategy in observational fear memory” (NCOMMS-22-34631A). We have found your comments and suggestions to be very helpful in improving our manuscript. We have extensively revised our manuscript and included additional experimental data that the reviewers had requested. We believe this extensive revision in both the figures and text has significantly improved our manuscript. You will find changes to the text highlighted throughout the revised manuscript, as well as cited by page and line number here. We have addressed the Reviewers’ overarching concerns as follows:

Reviewer #1 (Remarks to the Author):

In their manuscript, Terranova et al. explore what mice learn from witnessing other mice receive footshocks, and how this evolves over time after observation. In particular, two mice are placed in opposite compartments of a shuttle box, and the observer mouse on one side sees the demonstrator receive footshocks in the other. Thereafter, the observer mice are placed back into the observer compartment, the demonstrator compartment or a new context, either on day 1 after observation or on day 28. Behaviorally, the core finding is that mice freeze in the observer compartment all the time (day 1 & day 28), when placed in the demonstrator compartment, they freeze on day 28 but not day 1, when placed in a different compartment altogether (either a differently shaped box or the shuttle box with an opaque wall), they do not freeze at all. In agreement with the literature on memory, this suggests a widening of fear over time, with a representation very specific to where the mouse was when shocked on day 1 (observer compartment only), that becomes a bit wider (observer + similar demonstrator compartment but not the more different control box or opaque divider box) by day 28.

Neuroscientifically, arc positive cell counts suggest that the dorsal hippocampus is recruited when animals are replaced in the observer compartment 1d after shock observation, but not when replaced 28d later, with chemogenetic inhibition of the dorsal hippocampus interfering with freezing in observer compartment on day 1 but not day 28. In contrast, for the medial prefrontal cortex, deactivation using NMDA lesions, reduces freezing on day 28 but not day 1, and this is true also when deactivating the connections from mPFC to the BLA.

In contrast, when placed in the demonstrator compartment, arc was not increased in the dorsal hippocampus or mPFC on day 1, but was on day 28, significantly in the mPFC, and at trend level in the the dorsal hippocampus. mPFC->BLA inhibition triggered reduced freezing in the demo compartment on day 28.

While previous data had already shown that freezing on day 1 is stronger in the observation compartment than in a different context and other studies have shown the importance of mPFC, hippocampus and amygdala in this phenomenon, the current study provides interesting additions to the existing literature that are in line with the memory literature on contextual fear conditioning: that the specificity of the memory is reduced over time, and that that goes hand in hand with a reduced importance of the hippocampus. In addition, showing that the animals’ preference for the two compartments changes over time is interesting. However, there are several issues that would need to be addressed.

We thank the Reviewer for their support and feedback.

a) allocentric vs egocentric. The data presented in the paper shows that freezing that is specific to the exact context in which shocks were observed on day 1 generalizes to the similar demonstrator compartment context, but not to the very different control context or a box with an opaque divider. This 'gist'-like generalization to similar contexts over time is in line with the memory literature more generally. The authors however interpret it as reflecting a transition from egocentric to allocentric coding. These terms have very strong meanings in the visual neuroscience literature, and the authors do not sufficiently support the notion of an allocentric representation above and beyond the simpler concept of a gradual generalization to other, similar contexts. Framing the manuscript around these notions, seems misleading, as it suggests that there is proof of an allocentric representation suggesting a viewpoint independent but specific representation of space, that the data simply does not seem to support. We therefore recommend to remove these terms throughout, and replace them with more parsimonious terms: effects restricted to the observer compartment vs effects generalizing to the demonstrator compartment. Perhaps in the discussion, the authors can mention that this data could be compatible with a generalization gradient, in which the demonstrator compartment is more similar to the observer compartment than the other context or a box with opaque divider, or, alternatively, with the notion that the mPFC contains a representation of the entirety of the box, including both compartments, while the dHPC has a more fine grained representation restricted to the observer compartment, and that the importance of these two representational granularities switches over time, with behavior dominated by the finer-grained spatial representation in the dHPC on day1, and by the wider-scaled representation of the entire box on day 28. Still, a less fine grained representation is far from being 'allocentric'. Statements like "Egocentric and allocentric recall of observational CFC memory are fundamentally different 17 behavioral processes because, in egocentric recall, the observer recalls observational CFC memory 18 from their perspective, whereas, in allocentric recall, the observer recalls observational CFC 19 memory from the perspective of the demonstrator's context." really seem to go beyond the data: how do we know that the memory at play at d28 is from the demonstrators perspective?

We apologize for the confusing use of the terms "allocentric" and "egocentric" to describe the findings of our study. We agree with the Reviewer that it is necessary to use more parsimonious terms to describe the findings of our study more clearly, and to avoid going beyond the data. In the revised manuscript, as the Reviewer suggested, we removed all references to "allocentric" and "egocentric" throughout the text. We now directly say the behavioral manipulation that we did and the corresponding finding (e.g. "Recall of observational CFC memory in the observer chamber at the recent and remote time points positively correlates with observer freezing levels during observational CFC...", Page 3, Lines 29-30). We have updated the Introduction, Results, and Discussion to reflect these changes.

Regarding the neural mechanism of how observational CFC memory transforms over time to enable observers to recall this memory in both the observer and demonstrator chamber, we agree with the Reviewer that the mPFC may contain a wider-scaled representation of the observational CFC context, which encompasses both the observer and demonstrator chambers. Given that mPFC neural ensembles that encode direct fear experience can change over time during memory consolidation (Denardo et al., *Nature*, 2019), we hypothesized that, after systems consolidation of observational CFC memory, different subpopulations of mPFC neurons emerge that are associated with the observer or demonstrator chamber (Fig. 6). Consistent with our hypothesis, we found that, at the remote time point, there are multiple subpopulations of neurons in mPFC that are differentially activated by the observer or demonstrator chamber (Fig. 6F). We believe that, during the consolidation of the original observational CFC memory, a second subpopulation

of neurons emerges that is associated with demonstrator chamber. Furthermore, we speculate the demonstrator chamber-associated subpopulation of mPFC neurons may suggest the neural mechanisms for extraction of the knowledge that the demonstrator chamber is dangerous from the observational CFC episode, similar to what has been reported in humans (Sweegers et al., *Neuroimage*, 2014; Sweegers & Talamini, *Cortex*, 2014). We have updated the Discussion (Page 10, Line 17-40) to incorporate this idea. We thank the Reviewer for this thoughtful feedback, which has substantially strengthened our manuscript.

b) interactions: much of the statistics in the paper are restricted to t-tests, and the attribution of a particular function to a brain regions is based on the presence of an effect in one and an absence in another. This is statistically not the appropriate test for specificity: specificity corresponds to the demonstration of an interaction, i.e. that the effect is larger in one region or at one time point than another. The authors should thus replace the t-test with appropriate interaction tests (see Nieuwenhuis, S, B U Forstmann, and E.--J. Wagenmakers. "Erroneous Analyses of Interactions in Neuroscience: A Problem of Significance." *Nature Neuroscience* 14 (2011): 1105–7 for an explanation of this point). To illustrate some examples: Figure 1 (G-I) suggests that freezing is increased in the demonstrator compartment over time, by showing that the non-shock<Obs CFC at remote but not recent timepoints. What should be shown is timepoint x group interaction: is the group difference larger on d28 than d1. If this is not significant (it probably is, but if it isn't), it wouldn't be appropriate to suggest that the passing of time leads to an increase in freezing in the demo compartment. The same applies in Figure 2 for Arc in dorsal HPC, for the effect of chemogenetics on dHPC (larger in recent than remote rather than significant at recent but not remote), or Figure 3 and 4 when comparing remote and recent mPFC Arc expression, in the two compartments, Fig 5 when comparing the chemogenetic effects. In all these cases, conclusions can only be meaningfully drawn from these interaction effects. I understand that in some cases, the data may not be normally distributed, yet, ANOVAs are relatively permissive against modest deviations from their assumptions, which can be checked in a QQ plot. If data really is not normal, using a rank based method could be useful (as implemented in the CRAN package ARTool).

We thank the Reviewer for their feedback regarding how we can more properly conduct the statistical tests in our manuscript, for providing a useful reference, and for providing us with specific examples in our manuscript that we should address. In the revised manuscript, we have replaced the inappropriate use of t-tests with appropriate interaction tests by way of 2X2 between subjects or mixed ANOVAs, especially those suggested by the Reviewer, including the panels in Figure 1 (1B, 1E, 1G, 1I, 1O left, 1P left), Figure 2 (2D, 2G, 2K), Figure 3 (3C-F, 3I-J), Figure 4 (4B-4G, 4I), and Figure 5 (5J, 5K). In experiments in which there was only one time point tested (such as Fig. 1K, 1M; Fig. 5B-E) or experiments that served as confirmation of another finding (Fig. 1O right, Fig. 1P right) an independent t-test or Mann-Whitney U-test was used. In activity-dependent cell labeling experiments in which we examined reactivation of a neural subpopulation (Fig. 5N, 5P; 6C-F) a paired t-test was used. To directly compare reactivation of neural subpopulations between different groups in Figures 5 and 6, we conducted fold change analyses (Fig. 6G; Supplemental Fig. 4A). While there were a few instances where we failed to detect a difference as determined by the Bayesian factor (discussed below), the results of the interaction tests that we performed were consistent with our previous findings when we had used t-tests. We thank the Reviewer for this suggestion, which has strengthened the statistics and therefore the interpretations that can be drawn from the data in our manuscript. We have updated the Results throughout, Methods (Page 16, Lines 31-34), Figures, and Figure Legends to reflect these changes.

c) Evidence of absence: in several places, the manuscript interprets the lack of an effect (e.g. “There was no difference in Arc positivity between the observational CFC and the non-shock group in either dDG (Figure 4C, 4J) or dCA1 (Figure 4E, 4L) at the recent or remote time points”). To do so meaningfully, it is important to adjudicate whether the dependent variable provides evidence for H_0 vs an effect too weak to be significant given a specific group size. This should be addressed by reporting Bayes factors in addition to the p values (see Keyesers, Christian, Valeria Gazzola, and Eric-Jan Wagenmakers. “Using Bayes Factor Hypothesis Testing in Neuroscience to Establish Evidence of Absence.” *Nature Neuroscience* 23, no. 7 (July 29, 2020): 788–99. <https://doi.org/10.1038/s41593-020-0660-4>. For a tutorial on how to do this).

We thank the Reviewer for the suggestion on how to improve our interpretation on lack of effects, and for providing a helpful reference that describes how to calculate and interpret Bayesian factors. In the revised manuscript, we calculate Bayesian factors in addition to frequentist statistics using the opensource software JASP, and we report all Bayesian factors for all statistics that we conducted in Supplementary Table 1. For data with which we intend to claim no effect, we use Bayesian factors to determine if we indeed have evidence of absence or, alternatively, if our data are insufficient to determine if there is a lack of an effect. Our criteria for evidence of absence (i.e. evidence for the null hypothesis H_0), was $BF < 1/3$. While the Bayesian factors were mostly consistent with the p-values that we obtained, in situations where we found a non-significant p-value but a $BF > 1/3$ (such as in Fig. 1G, Fig. 1O right, Fig. 3F, Fig. 5B, Fig. 5E) we state in the Results section that “we could not find a difference” for the particular comparison. We have updated the Methods section with a description of how we calculated and interpreted the Bayesian factors, particularly with regards to claiming lack of an effect (Page 16, Line 38-42). We have also updated the Results throughout consider the Bayesian factors in our interpretation of data in which we claim a lack of an effect.

d) the group sizes are sometimes 10, sometimes 15, and it is unclear why such fluctuations exist. The methods section does not sufficiently explain these choices: “Statistical methods were not used to predetermine sample sizes in experiments; sample sizes were based on what is conventional for the field and previous studies”. Why previous experiments would suggest $N=9,10$ or 15 based on different experiments is unclear.

We apologize for the lack of clarity regarding the rationale for the sample sizes, particularly in Figure 1. In previous studies, including our own studies, it was determined that enough statistical power could be achieved by having a minimum sample size of $N = 10$ per group in behavioral experiments. Therefore, in the revised manuscript, we ensured that all behavioral experiments had a sample size of at least $N = 10$ per group. In some experiments in Figure 1 in which we had $N = 15$ (such as in Fig. 1A-I), we had conducted an experiment with an additional cohort $N = 5$ mice beyond the original cohort of mice $N = 10$ mice that we ran, to verify a novel experimental findings that were crucial to our manuscript (such as higher freezing levels compared with the non-shock group at the remote time point for observers subjected to recall in the observer or demonstrator chamber, Fig. 1B; Fig. 1I). We agree with the Reviewer that there is not a clear rationale to have some groups in Fig. 1 with an $N = 15$. Therefore, to be consistent with other experiments, we removed the additional cohort of $N = 5$ mice in the remote time point groups in Figure 1A-I, so now all $Ns = 10$ per group. Indeed, even with Ns reduced to 10, we have enough statistical power to detect differences in the effect of Obs CFC on recall of observational CFC memory at the remote time point in the observer chamber (Fig. 1B) and demonstrator chamber (Fig. 1I). Regarding the optogenetics experiments in Figure 3I, because we were concerned that the surgical and testing procedures may produce additional variability in observer freezing levels,

some groups have Ns higher than 10. In the revised manuscript, we have updated the Methods (Page 16, Lines 26-29) to describe the rationale more clearly for the sample sizes that we selected. We have also included Supplementary Table 1, which contains all sample sizes and statistics for all experiments.

Minor comments:

Figures: please add dots for each individual, and plot freezing always on the same scale for all experiments so that readers can judge the relative levels of freezing across the many experiments, and judge whether 'control' conditions in various cohorts really produced 'typical' levels of freezing. Otherwise, some control vs experimental differences could be driven by differences in the control condition.

As the Reviewer has suggested, we have added dots for each individual data point on each graph throughout the manuscript. For the behavioral experiments, we have plotted the data with a y-axis scale of 30% freezing time. Exceptions to this are as follows: In the chamber preference experiments Fig. 1O (left) and 1P (left), we have plotted the data with a y-axis scale of 15% freezing time because the overall testing time for this experiment (12 minutes) is different than all other experiments (4 minutes), and the testing chamber in this experiment is different as well. Because the total testing time is different, it would be misleading to encourage direct comparison of these graphs (1O left and 1P left) with graphs from other behavioral experiments using the same y-axis scale. In graphs where individual data points exceed 30%, such as the optogenetics experiments in Fig. 3I and 4I, we plotted these graphs with a y-axis scale of 60% freezing time in order to show all data points.

P5, l43: 'silencing of dHPC' should probably be mPFC.

We corrected the typo.

Fig4, panel A, Mouse should probably be in the demo compartment also on day 1.

We have corrected the figure panel and moved the mouse to the demonstrator chamber.

In the various figures, please replace absolute freezing time with %-freezing to facilitate comparison with other studies that may have monitored freezing over different durations.

We have changed the absolute freezing time to % freezing time in all the Figures, as the Reviewer has suggested.

Line 6, page 9 completion rather than competition likely

We corrected the typo.

For Place preference test, animals always placed in demos chamber?

Yes, we initially placed all observers into the demonstrator chamber for the place preference test. We updated the Methods (Page 12, Lines 25-26) sections to clearly indicate this protocol.

Fig5: In legend 'J' and 'H' should be interchanged.

We have corrected the typo.

P5I2 'prior to egocentric recall'. It would be good to provide more details about the timing of the lesion relative to shock observation and recall either in the text of figure 3.

Since we have replaced the lesion experiment with additional optogenetics experiments in the revised manuscript (see R2#4 in Major Comments below), we have removed this text and other text corresponding to the lesion experiment from the Methods.

P6: "at both the recent and remote time point". Add s at time points.

We have corrected the typo.

P7: "These results indicate that after the systems consolidation, observers acquire the ability to allocentrically recall observational CFC memory. ". This is an example of what needs to be reframed: generalization from the observer to the demonstrator compartment is a parsimonious explanation of the same data.

We thank the Reviewer for pointing out this example in the text. As stated above in R1#1 in Major Comments, in the revised manuscript, we removed all references to "allocentric" and "egocentric" throughout the text. Instead, we now directly say the behavioral manipulation that we did and the corresponding finding.

P8: "divided by a transparent plexiglass partition, as previously described ". Was the divider also perforated?

The divider was not perforated in the current manuscript, nor in the previous manuscript (Terranova et al., *Neuron*, 2022). We have clarified this point in the Methods (Page 11, Lines 40-41).

P11: "NMDA-Induced Excitotoxic Lesion mPFC was bilaterally injected with 150 nL per side of NMDA (25 mg/mL) or saline (control) aimed at the following coordinates relative to bregma: AP: +2.00 mm, ML: ± 0.30 mm, DV: -2.00 29 mm. Observers recovered for 1 week before being group housed with cagemates". Still not quite clear to me, whether lesions were made prior to shock observation, or after observation but before recall? And how long before shock observation was the NMDA applied?

Since we have replaced the lesion experiment with additional optogenetics experiments in the revised manuscript (see R2#4 in Major Comments below), we have removed this text and other text corresponding to the lesion experiment from the Methods.

P11: “13 seconds of laser stimulation on and 2 seconds of laser stimulation off”: was a rebound observed behaviorally during 2s off?

As the Reviewer suggested, we examined whether there was a rebound effect in the ArchT mice during the 2-second Light-Off period in recall of observational CFC memory in the observer chamber or demonstrator. We found that there was no difference in observer freezing levels in the Light-On vs. Light-Off periods during recall of observational CFC memory in the observer chamber (Supplementary Fig. 3A) or demonstrator chamber (Supplementary Fig. 3B).

Fig1: A Mann-Whitney U is used for A-C, but a student t-test for some of the other tests. Panel J-K, No recall in novel environment is a critical test, and yet $P=0.07$. Here also the difference in N becomes intriguing, as the same effect size for $N=15$ would be significant.

As the Reviewer suggested above in R1#2 in Major Comments, we replaced the statistical tests with interaction tests by two-way ANOVA, so we removed the Mann-Whitney U-test from Fig. 1A-C. The revised manuscript has our rationale for choosing to conduct a Mann-Whitney U-test instead of a t-test in Methods (Page 16, Line 34-36). Regarding the data in Fig. 1J-K, we agree that no recall in a novel environment is a critical test. To further address this point, especially given the concerns raised in R2#3 in Major Comments and R4#1 in Major Comments, we performed an additional novel context experiment. In this experiment, observers were subjected to observational CFC (or non-shock control) and, 28 days later at the remote time point, were introduced to a new context that was the same as the demonstrator except that the shock grid was covered with an opaque floor (Fig. 1L). Even in this very similar context, observer freezing level was the same between observational CFC and non-shock groups (Fig. 1M). Thus, observers are not exhibiting fear generalization during recall of observational CFC in the demonstrator chamber.

Fig2: include a non-shock arc/NeuN micrograph for comparison. Also specify how arc% is calculated (double positive count (Arc&NeuN)/(NeuN)?

As the Reviewer suggested, we have updated Fig. 2 to include non-shock Arc/NeuN example pictures. We have also specified how Arc % is calculated in the Methods (Page 13, Lines 45-46; Page 14, Line 1).

Fig3G: the Sal group clearly contained an outlier. Recalculate U value without the outlier.

Since we have replaced the lesion experiment with additional optogenetics experiments in the revised manuscript (see R2#4 in Major Comments below), we have removed this figure and the corresponding text from the manuscript.

Fig4: again include no-shock micrograph for comparison.

As the Reviewer suggested, we have updated Fig. 3 (original manuscript Fig. 4) to include non-shock Arc/NeuN example pictures.

Fig4N: here the non-shock seems low compared to other panels, rather than obsCFC increased.

We agree with the Reviewer that the percentage of Arc positive neurons seemed lower in the non-shock group compared with other panels, such as the recent time point data in Figure 4G of the original manuscript. However, because we conducted the experiments for Figure 4N and 4G of the original manuscript at different times, we were unable to directly compare the Arc levels between these data. Therefore, to examine the consistency of Arc positivity in the non-shock group in Figure 4N, we repeated the experiments in Figure 4N and 4G of the original manuscript, so that we could make direct comparisons. We found that Arc positivity in the non-shock groups at both recent and remote was consistent across mPFC subregions (4D-4G). We have updated the data in the Results (Page 6, Lines 8-16) and Figures.

Fig5K: in the mcherry group, freezing should be similar to Fig 1I obs CFC. Is that the case? This also emphasizes the importance of having all freezing graphs on the same scale.

We did not directly compare the data in Figure 5K with those in Figure 1I because the experimental conditions are very different between these figures (i.e. observers in Fig. 5K have surgery, viral injection, and multiple days of testing whereas observers in Fig. 1I do not have these). However, we agree with the Reviewer that it is better to present all the data using the same scale. As the Reviewer suggested in R1#1 and R1#4 in Minor Comments, we have adjusted the data to have the same y-axis scale and to be expressed as a percentage.

Reviewer #2 (Remarks to the Author):

Terranova et al. addressed the characteristics and neural mechanisms underlying observational CFC in one's own or a party's point of view, discovering that a certain amount of time must pass before allocentric recall becomes possible and that mPFC (and BLA) is involved in the recall. Although it is unclear now do the authors define mPFC (does it include not only PL/IL but also cingulate cortex or ACC?), this study introduces a new perspective on observational [fear] learning. Some findings are still primitive, but they are highly suggestive. My comments are as follows:

We thank the Reviewer for their support and feedback.

Major comments

1. About freezing in the Dem chamber under the Remote conditions (perhaps the most important finding herein). I would like to see not only the references of the upstream brain regions involved in egocentric and allocentric recalls, but also a discussion of why it takes so long to acquire the defensive behavior. Although this may be a general question in the systems consolidation, it was strange for me that it did not freeze in Dem Recent. If the authors assume that the shift of Dem Recent to Dem Remote is not a generalization, I am even more curious about the mechanism.

We thank the Reviewer for this comment, and agree that this is an interesting point. In the revised manuscript, using an activity-dependent cell labeling strategy, we examined the subpopulation of neurons in the prelimbic subregion of the mPFC that were activated during recall of observational CFC memory in the observer and demonstrator chambers at the remote time point (Fig. 6). We found that, at the remote time point, there are multiple subpopulations of neurons that are

differentially activated by the observer or demonstrator chamber (Fig. 6F). We believe that, during the consolidation of the original observational CFC memory, a second subpopulation of neurons emerges that is associated with demonstrator chamber. Furthermore, we speculate the demonstrator chamber-associated subpopulation of mPFC neurons may represent an extraction of the knowledge that demonstrator chamber is dangerous from the observational CFC experience, similar to what has been previously reported in humans (Sweegers et al., *Neuroimage*, 2014; Sweegers & Talamini, *Cortex*, 2014). Since this process is occurring during systems consolidation of the original observational CFC memory episode, we think that this explains why it takes observers so long for mice to obtain the knowledge that the demonstrator chamber is dangerous as a semantic memory. We have updated the Results (Page 7, Lines 27-46; Page 8 Lines 1-7), Discussion (Page 10, Lines 17-40), Figures, and Figure legends to consider this mechanism.

2. Regarding the consideration of CS-dependent (Result Page.3, L27-28). If it is CS-dependent, then elimination learning should be possible. What do the authors think is the case? Is there a difference in the freezing percentage between the early and late stages during the Recall test? More specifically, in general contextual fear conditioning, a temporal transition of increasing and decreasing freezing is observed. Did the authors see such a time-dependent phenomenon? Even if not, I would like to see a time change in the freezing percentage, rather than the total freezing time. This data would be an indicator of the type of information and timing to which mice exhibit freezing.

We agree with the Reviewer that elimination learning should be possible. As the Reviewer suggested, we examined the kinetics of the observer freezing response during recall of observational CFC in the observer chamber (Fig. 1E) and demonstrator chamber (Supplementary Fig. 2C), and did not find difference in observer freezing levels throughout the testing time. As stated above in R1#1 Minor Comments, in the revised manuscript, we have expressed the data as a percentage of freezing rather than total freezing time.

3. On the denial of the possibility of generalization (Result L39-44). I think that Fig.1J,K are important data. I was concerned about "scented with 1% acetic acid" (Methods Page 9, L 22). I think the author probably employed a common method, but I am wondering if the acetic acid itself might be a stimulus to increase activity (search behavior). I think that there is probably a floor effect, with $P = 0.07$, but I am more surprised that the Obs CFC group tends to have a lower freezing rate.

We thank the Reviewer for raising this issue, and agree that the data in Figure 1J-K are important. As stated in R1#14 in Minor Comments above, we performed an additional novel context experiment. In this experiment, observers were subjected to observational CFC (or non-shock control) and, 28 days later at the remote time point, were introduced to a new context that was the same as the demonstrator except that the shock grid was covered with an opaque floor (Fig. 1L). Importantly, there was no scent added to this context. Even in this context that was very similar to the demonstrator chamber, there was no difference in observer freezing level between observational CFC and non-shock groups (Fig. 1M). Thus, observers are not exhibiting fear generalization during recall of observational CFC in the demonstrator chamber.

4. Fig.3 F; It is difficult to evaluate if only mPFC is lesioned. It is necessary to check the leak to see if the lesion affected other important areas such as ACC. I think that even in the representative image in Fig 5E, the injected area is too large to claim to be specific, and the authors cannot rule

out that the data obtained in this study reflected indirect effects through other brain areas. The authors must convincingly prove the specificity of their neural manipulation. Since the roles of PL, IL, and ACC are very different, it is essential to distinguish these brain regions properly.

We apologize for the lack of clarity in Figure 3F of the original manuscript regarding the mPFC lesion. Since the lesion experiment was a duplication of optogenetics experiment, we have removed the lesion experiment data from the revised manuscript and replaced it with additional optogenetics data. Specifically, we added Light-Off control groups in Fig. 3I-J (as suggested in R3#4 below), increased the N in Fig. 4I, and assessed the possibility of a “rebound effect” in observer freezing levels during the 2-second Light-Off period in Supplemental Fig. 3 (as suggested by R1#13 in Minor Comments above).

We agree with the Reviewer that the roles of the PL, IL, and ACC are very different and that it is critical to properly distinguish between these brain regions. In the revised manuscript, using the boundaries defined by the Allen Institute Brain Atlas, we analyzed Arc expression in the secondary motor cortex (SOM), ACC, PL, and IL after recall of observational CFC memory in the observer chamber (Fig. 3C-F), recall of observational CFC memory in the demonstrator chamber (Fig. 4D-G), and after exposure to observational CFC (Fig. 5B-E). By combining analysis of these data with a more in-depth consideration of the literature, we have expanded the Discussion (Page 8, Line 31-46; Page 9 Line 1) to consider the contribution of these mPFC subregions to the regulation of observational CFC memory formation and recall more thoroughly. We thank the Reviewer for raising this point, as it has allowed us to ascertain a more refined viewpoint of how the mPFC regulates recall of observational CFC memory.

5. Poeple have questioned whether ArchT can really suppress the activity of axon terminals. Although ArchT can suppress the generation of action potentials, in general, it does not have such a strong effect that it suppresses action potentials once generated. The authors have used this method in their previous reports too, but they should demonstrate that ArchT indeed works under their in vivo experimental conditions. Otherwise, this work will again be viewed with skepticism by researchers in the same field, as was the case in the previous reports.

We thank the Reviewer for raising this potential issue, and we share this concern regarding ArchT. When determining our stimulation protocol for terminal inhibition, we first considered that continuous stimulation of ArchT at the terminals (> 40 seconds) can produce opsin-independent effects that can lead to spontaneous vesicle release and blunt terminal inhibition (Mahn et al., *Nat Neurosci*, 2016). Previous reports such as (McHugh et al., *Nat Neurosci*, 2022) have demonstrated that a repeated Light-On / Light-Off strategy throughout a behavior testing period still produces potent terminal inhibition while avoiding problems associated with continuous stimulation of ArchT. Therefore, we adopted a 13-second light-On / 2-second light-Off protocol to avoid these opsin-independent effects of ArchT while still inhibiting mPFC terminals in BLA during the observational CFC memory recall test.

While we did not directly examine the ability of ArchT to suppress action potentials that have already been generated in our manuscript, we have obtained several lines of evidence that suggest that our ArchT manipulations were sufficient to inhibit the neural activity to recall observational CFC memory by observers. First, we have included a Light-Off control group, as suggested by R3#4 below, in which mice never receive laser stimulation to activate ArchT in infected neurons. We demonstrate that both Light-On + ArchT are required to achieve inhibition of observation freezing response during recall of Observational CFC memory at the remote time

point (Fig. 3I), indicating that ArchT stimulation is sufficient to reduce observer freezing levels in recall of observational CFC memory. Next, after the observational CFC recall test in the optogenetics experiment, we examined Arc positivity in BLA in four groups: Light-Off + eYFP, Light-Off + ArchT, Light-On + eYFP, Light-On + ArchT. Consistent with our findings in Fig. 3I, Light-On + ArchT mice had reduced Arc-positivity in BLA compared with all other groups (Fig. 3J), suggesting that stimulation of ArchT is sufficient to inhibit recall of observational CFC memory by observers via reducing mPFC-BLA neural activity. Finally, as suggested by R1#13 in Minor Comments above, we examined the possibility of a “rebound effect” in observer freezing levels during the 2-second Light-Off period and found that observer freezing response was still suppressed even during the Light-Off period (Supplemental Fig. 3), suggesting that stimulation of ArchT is sufficient to continue to inhibit recall of observational CFC memory even in the Light-Off period. Together, these lines of evidence suggest that Light-On + ArchT is sufficient to inhibit mPFC terminals in BLA during recall of observational CFC memory. We thank the Reviewer for raising this concern, and we have updated the Results (Page 6, Lines 16-21), Methods (Page 15, Lines 18-20), Figures, and Figure Legends with these additional data.

6. In the Discussion (P7L28-30), the authors described " We also found that the activation of subpopulation of mPFC neurons during observational CFC is necessary for both egocentric and allocentric recall of the remote observational CFC memory (Figure 5H, 5K), suggesting that same population of mPFC neurons may regulate both egocentric and allocentric remote recall process". In Fig. 5, as in Fig. 3-4, by manipulating only the cells that project to BLA, the author can show that the "same" cell population is involved.

We thank the Reviewer for the suggested experiment to help answer this important question. Based on this suggestion and comments from other Reviewers, in the revised manuscript, we employed an activity-dependent cell labeling strategy in the TRE-Cre mouse line to label the subpopulation of mPFC neurons activated during Obs CFC (for description see Results, Page 7, Line 5-14). This allowed us to examine the reactivation of this subpopulation in the observer or demonstrator chamber at the remote time point. We found that the subpopulation of mPFC neurons activated during observational CFC were reactivated during recall of observational CFC memory in the observer chamber (Fig. 5N) or demonstrator chamber (Fig. 5P). We have updated the Results (Page 7, Lines 5-25), Discussion (Page 10, Lines 11-15), and Figures, and Supplemental Figures to consider these data.

Since recall of observational CFC memory in the observer and demonstrator chambers should be fundamentally different behavioral processes, we hypothesized that the neural mechanisms that regulate recall of observational CFC memory in the observer or demonstrator chamber would diverge. Therefore, we used an activity-dependent cell labeling strategy to examine the subpopulation of neurons in the prelimbic subregion of the mPFC that were activated during recall of observational CFC memory in the observer and demonstrator chambers at the remote time point (Fig. 6). We found that, at the remote time point, there are multiple subpopulations of neurons that are differentially activated by the observer or demonstrator chamber (Fig. 6F). We believe that, during the consolidation of the original observational CFC memory, a second subpopulation of neurons emerges that is associated with demonstrator chamber. We thank the Reviewer for proposing this very helpful experiment, which helped us to significantly improve the quality of our manuscript.

Minor comments

1. What is the reason for the difference in experimental conditions? 30-45min (Methods page 9; L 7-8) vs 5-40min (L14). Typo?

We thank the Reviewer for mentioning this point. This is not a typo, it is the procedure that we followed. Because observational CFC is a stressful procedure for observers, to mitigate the effects social transfer of fear, observers are maintained in empty holding cages 30-45 minutes after the conclusion of observational CFC test before returning to their respective home cages. Since initial separation of observers prior to observational CFC or the observational CFC memory recall test is much less stressful, we do not need to wait as long before testing. Importantly, all observers, regardless of experimental manipulation or group, are subjected to the separation procedures before and after behavior testing. Furthermore, we have optimized and effectively used these procedures in our previous study (Terranova et al., *Neuron*, 2022). We have updated the text in the Methods (Page 12, Lines 3-6) to clarify this point.

2. hM4Di is incorrectly described as AAV2/8 (Methods page 9; L 36-37) and mCherry as AAV2/5 (Page 11, L17-18)? The text and figure are correct.

We have corrected the typos in the Methods.

3. The number of Arcs also seems to increase outside of mPFC ROI. Why? Please quantify them.

As the Reviewer has suggested, we have quantified Arc in the secondary motor cortex (SOM), which is the region outside the mPFC ROI where Arc seemed to increase in Fig. 3C, 4D, and 5B. We found that Arc-positivity in SOM is higher in the Obs CFC group compared with the non-shock group during recall of observational CFC memory in the observer chamber at the remote time point (Fig. 3C). We did not detect any differences in SOM Arc-positivity in the demonstrator chamber (Fig. 4D) or during Obs CFC exposure (Fig. 5B). We consider these data in the Discussion (Page 8, Lines 31-44).

4. Fig. 4A; Is the position of the mouse a drawing error? Since it's Allocentric, shouldn't the mouse be on the grid?

We thank the Reviewer for pointing out this error, and apologize for the incorrect placement of the observer mouse in this cartoon. We have moved the mouse to the demonstrator chamber.

5. Some graphical abstract would help the readers.

We agree that a graphical abstract would be helpful to include. The revised manuscript now contains a new figure (Fig. 7), which provides a graphical summary of the main findings of this manuscript.

6. I would like to see some discussion on the difference between mPFC and ACC in terms of their contribution to observational CFC, since there is no mention of ACC in the Discussion or the Results although ACC is included in the Introduction (P1L43).

We thank the Reviewer for this suggestion, and have expanded the Discussion (Page 9, Lines 1-7) to further consider differences between the mPFC and ACC in their contributions to observational CFC.

7. I do not understand why the author used NMDA-induced excitotoxicity instead of DREADD for the suppression of mPFC (Fig. 3). Is it because it's easy?

As stated above in R2#4 in Major Comments, we have replaced with lesion experiments in the manuscript with additional optogenetics experiments.

Reviewer #3 (Remarks to the Author):

The manuscript by Terranova et al. seeks to study the underlying neural mechanisms that contribute to empathic observational CFC memory. They provide data showing that observational CFC training can be consolidated into long-term memory and found different brain areas are responsible for either egocentric or allocentric retrieval of recent or remote observational CFC memory. The authors used a large variety of techniques to address their question, however, there are still a number of issues with the significance and approach of the paper.

We thank the Reviewer for their support and feedback.

1. During the observational CFC memory training session, what's the freezing level of the observer mice when demonstrator mice experience strong and lasting foot shocks? Is the freezing level or duration of gazing at the opposite chamber correlated with the recall performance afterward?

We thank the Reviewer for raising these questions. We have previously determined in our recent manuscript, (Terranova et al., *Neuron*, 2022), that the average freezing levels of observers when the demonstrator receives strong and lasting foot shocks is ~30% of the shock period (Fig. 1E in the *Neuron* manuscript). Regarding the kinetics of the observer freezing response (Fig. 1F in the *Neuron* manuscript), observers robustly freeze after shock delivery to the demonstrator, and this freezing response is long-lasting (albeit declining slightly over time during the shock interval). When we use an opaque partition, so the observer cannot see the demonstrator, we suppress observer's freezing response, indicating that observational CFC requires a visual perception stimulus (Fig. 1E-F in the *Neuron* manuscript). For your convenience, we have adapted these data from the original *Neuron* manuscript and included them below:

[redacted]

(Adapted from Terranova et al., *Neuron*, 2022. Middle panel: Shaded box; demonstrator shock period. Right Panel: T; transparent partition. O; opaque partition)

In the revised manuscript, we quantified the freezing levels of observers during Observational CFC, and then correlated these freezing levels with observer freezing levels during the recall test in the observer chamber or demonstrator chamber. We found that observer freezing levels during observational CFC correlates with freezing levels during recall of observational CFC memory in the observer chamber (Fig. 1C-D) but not the demonstrator chamber (Supplemental Fig. 2A-B). We describe these data in the Results (Page 3, Lines 29-32; Page 4, Lines 8-11), Figures, and Supplemental Figures.

2. Head direction is pivotal to egocentric spatial encoding from the observer's perspective. What's the main head direction of observer mice behaving freeze during egocentric or allocentric recall of CFeC?

We thank the Reviewer for raising this point, and agree that head direction is critical to examine. In the revised manuscript, we quantified the head direction in 4 directions during the recall test in the observer chamber (Supplemental Fig. 1A) or demonstrator chamber (Supplemental Fig. 1C). We found that observer head direction is preferentially oriented towards the demonstrator chamber during recall of observational CFC memory in the observer chamber (Supplemental Fig. 1B). In contrast, there is no preferred head direction for observers during recall of observational CFC memory in the demonstrator chamber at the remote time point (Supplemental Fig. 1D). We describe these data in the Results (Page 3, Lines 35-41; Page 4, Lines 11-13).

3. In Fig2N, why the freezing time in remote recall (both mcherry and hm4d treated mice) are longer than the recent recall (Fig1 C and Fig 2M)?

We thank the Reviewer for pointing this out, and we also wondered why the freezing levels were higher in Figure 2N than in Figure 1C or Figure 2M of the original manuscript. At first, we thought this may have happened by chance, so we repeated this experiment using the same injection and testing protocols. However, we obtained a similar result, with higher freezing levels in the mCherry and hm4Di groups at the remote time point. We considered that the duration of incubation time of the virus in dHPC may somehow contribute to the differences in the behavior that we are seeing (3 weeks at the recent time point and 7 weeks at the remote time point). Therefore, in the revised manuscript, we repeated the remote time point dHPC DREADD experiment, and adjusted our experimental schedules so that the viral incubation times for the recent time point and remote time point groups were the same (Fig. 2I-J). By reducing the viral incubation times for the observers in the remote time point groups, we were able to obtain freezing levels that were consistent with the recent time point groups (Fig. 2K)

4. Fig2 lacks Arc staining of no-shock group mice. Fig3K lacks the BLA Arc level after laser stimulation in the no-shock group. Light-off control experiments are also needed.

We apologize for lack of images in control groups in Figures 2, and have provided these images in the revised manuscript. We also agree with the Reviewer that it is important to examine the Light-Off condition. We have added the Light-Off condition in both the behavioral (Fig. 3I) and Arc in BLA (Fig. 3J) experiments for the optogenetics experiments the Arc experiments. These data improve the interpretation of our optogenetics data, and help to demonstrate the efficacy of the ArchT manipulation we used to inhibit mPFC terminals in BLA. We thank the Reviewer for this helpful suggestion.

5. Fig4A, placement of Obs mice in recall test seems wrong? It should be placed at the Dem chamber.

We apologize for the incorrect placement of the observer mouse in this cartoon, and have moved it to the demonstrator chamber.

6. Page5 Line43 dHPC should be corrected by mPFC.

We have corrected the typo in the Discussion.

7. What impresses me most, in this study, is the neural circuit underlying observational CFC is almost the same with that of well-studied direct CFC memory recall. Maybe specific subpopulations of cells or projections from the same brain areas, or specific in-vivo firing patterns, contribute to the formation of observational CFC and to the perspective-dependent and time-dependent retrieval of observational CFC memory. It is a pity that the authors did not solve these problems any further.

We agree with the Reviewer that this is a crucial observation, and that it is important to further interrogate how observational CFC memory is formed and recalled. First, as stated in R2#6 in Major Comments above, we employed an activity-dependent cell labeling strategy in the TRE-Cre mouse line to label the subpopulation of mPFC neurons activated during exposure to Obs CFC (for description see Results, Page 7, Line 5-14), which allowed us to examine the reactivation of this subpopulation in the observer or demonstrator chamber at the remote time point. We found that the subpopulation of mPFC neurons activated during exposure to observational CFC were reactivated during recall of observational CFC memory in the observer chamber (Fig. 5N) or demonstrator chamber (Fig. 5P).

Next, using an activity-dependent cell labeling strategy, we examined the subpopulation of neurons in the prelimbic subregion of the mPFC that were activated during recall of observational CFC memory in the observer and demonstrator chambers at the remote time point (Fig. 6). We found that, at the remote time point, there are multiple subpopulations of neurons that are differentially activated by the observer or demonstrator chamber (Fig. 6F). We believe that, during the consolidation of the original observational CFC memory, a second subpopulation of neurons emerges that is associated with demonstrator chamber. Furthermore, we speculate the demonstrator chamber-associated subpopulation of mPFC neurons suggest the neural mechanisms for extraction of the knowledge that the demonstrator chamber is dangerous from the observational CFC episode, similar to what has been reported in humans (Sweegers et al., *Neuroimage*, 2014; Sweegers & Talamini, *Cortex*, 2014). We thank the Reviewer for this feedback, which has significantly improved our manuscript.

Reviewer #4 (Remarks to the Author):

Memories for events initially depend on the hippocampus, but with time become increasingly dependent on extra-hippocampal regions for their expression. This time-dependent process of reorganization is known as systems consolidation, and has been especially characterized in rodent models using contextual fear conditioning tasks. The current manuscript by Terranova and colleagues examines whether systems consolidation occurs for another form of memory—

observational fear memory—where rodents learn about environmental contingencies through observation of conspecifics. A particularly interesting aspect of this task is the potential to distinguish between ego- and allocentric recall (i.e., recall from the subject mouse's perspective vs. the demonstrator's mouse perspective). The work presented here shows that observational fear memories may be recalled egocentrically at recent and remote delays. However, they also make a case that observational fear memories may be recalled allocentrically at remote delays, suggesting there is some additional 'insight' that emerges with time and may track time-dependent changes in memory organization. With respect to these time-dependent changes, using IEG analyses and chemogenetic regional activation/silencing approaches they establish that recall of recent memories engages dorsal hippocampus, whereas remote recall engages mPFC-BLA projections. Additionally, they show that mPFC activity, at the time of encoding, is necessary for the formation of remote observational fear memories. This work is an important addition to the literature on systems consolidation, providing further support for a time-dependent shift from hippocampus to cortex in a novel learning paradigm. The experiments are carefully conducted, and the manuscript is well-written. That said, we identified two important issues that could be addressed in a revision.

We thank the reviewer for their support and feedback.

1. Memory generalization vs. allocentric observational fear. In experiment presented in Fig. 1G-, the authors found that observer mice freeze in the demonstrator's chamber only at remote time-point, and this is interpreted as an example of allocentric observational fear. An alternative possibility, as acknowledged by the authors, is that this reflects memory generalization. In another words, the fear associated to the observer chamber can generalize to demonstrator chamber as the memory ages. To address this, authors exposed observational CFC mice to a novel context (Fig. 1J-K) and concluded absence of generalization due to low freezing at novel context. However, we have the following concern. Perhaps the novel context is very different, and generalization might be observed in a more similar context. Since the observer chamber and demonstrator chamber share so many features (especially that observer mice can see the demonstrator context) it might be the case that the demonstrator chamber is functioning as a similar context. Moreover, previous exposure to a context can accelerate generalization to it. The fact that observer mice are exposed to demonstrator context therefore could lead to a faster generalization of the fear memory for this context compared to a novel context. Finally, and most importantly, in experiment 1D, authors have shown that an opaque wall during the recall impedes the recall of the observational CFC. This outcome indicates that some features of the demonstrator chamber are being used for recall, so it would be much easier for mice to generalize the fear to this context rather than a novel context. One potential solution to distinguish allocentric recall from a generalized recall would be to add a third chamber. This way at the same time the experiment would include a no-shock demonstrator in chamber A, observer mouse in chamber B (at the center so the observer can see both of the demonstrators), and a shocked demonstrator in chamber C. At the time of recall the observer would be exposed to chamber A. If freezing level elevates this means the observer could not simply distinguish between the chambers and hence the memory was generalized. On the other hand, if the observer doesn't freeze in chamber A (while freezing to chamber C), then the conclusion is the mouse can distinguish the chambers and it only freezes in an allocentric situation. We appreciate that this type of experiment might not be possible, but we are at least interested in the authors' thoughts on this issue.

We appreciate the Reviewer's thoughtful consideration about the distinction between memory generalization and other's context-focused recall of observational CFC memory. We agree with the Reviewer that the new context is very different than the context where observers were

subjected to observational CFC. As stated above in R1#14 in Minor Comments and R2#3 in Major Comments above, we performed an additional novel context experiment. In this experiment, observers were subjected to observational CFC (or non-shock control) and, 28 days later at the remote time point, were introduced to a new context that was the same as the demonstrator except that the shock grid was covered with an opaque floor (Fig. 1L). Even in this very similar context, observer freezing level was the same between observational CFC and non-shock groups (Fig. 1M). Thus, observers are not exhibiting fear generalization during recall of observational CFC in the demonstrator chamber. Instead, their recall of observational CFC memory in the demonstrator chamber seems to be specifically triggered by the shock grid floor.

2. Global activation/inhibition vs. engram tagging. The authors chemogenetically activated/silenced entire regions. The application engram cell specific manipulations would be interesting, especially with respect to the possibility to distinguish between ensembles encoding ego- vs. allocentric memory in the mPFC.

We agree with the Reviewer that it would be interesting to apply engram cell-specific manipulations to examine the role of mPFC in recall of observational CFC memory in the observer or demonstrator chambers. Therefore, we conducted two experiments to examine this, as stated in R2#1, R2#6 in Major Comments and R3#7 above. First, we employed an activity-dependent cell labeling strategy in the TRE-Cre mouse line to label the subpopulation of mPFC neurons activated during exposure to Obs CFC (for description see Results, Page 7, Line 5-14), which allowed us to examine the reactivation of this subpopulation in the observer or demonstrator chamber at the remote time point. We found that the subpopulation of mPFC neurons activated during exposure to observational CFC were reactivated during recall of observational CFC memory in the observer chamber (Fig. 5N) or demonstrator chamber (Fig. 5P).

Next, using an activity-dependent cell labeling strategy, we examined the subpopulation of neurons in the prelimbic subregion of the mPFC that were activated during recall of observational CFC memory in the observer and demonstrator chambers at the remote time point (Fig. 6). We found that, at the remote time point, there are multiple subpopulations of neurons that are differentially activated by the observer or demonstrator chamber (Fig. 6F). We believe that, during the consolidation of the original observational CFC memory, a second subpopulation of neurons emerges that is associated with demonstrator chamber. Furthermore, we speculate the demonstrator chamber-associated subpopulation of mPFC neurons suggest the neural mechanisms for extraction of the knowledge that the demonstrator chamber is dangerous from the observational CFC episode, similar to what has been reported in humans (Sweegers et al., *Neuroimage*, 2014; Sweegers & Talamini, *Cortex*, 2014). Figures, and Figure legends to consider this mechanism. We thank the Reviewer for raising this issue, which has greatly strengthened our manuscript.

Minor

1- The freezing levels for some cases are reported in seconds. It would be better to report them all in percentages.

We agree with the Reviewer that it is better to report freezing levels as percentages. In the revised manuscript, all freezing levels are reported as percentages.

2- Overall, the freezing levels are pretty low (around 10-20 seconds, or 4-8% in case of 4 min

recalls). Authors should be cautious about how meaningful a 4% change in freezing could be for their behavior paradigm.

We agree with the Reviewer about the low freezing levels, and have been careful in the revised manuscript not to overinterpret our data. Moreover, the revised manuscript now uses interaction tests and Bayesian factors, based on Reviewer #1's comments, to help improve the interpretation of our data.

3- The rationale for using Mann-Whitney U-test for the first comparison of freezing levels (used only for fig 1.A-C) should be provided (t-tests were used for remaining figures)..

Because we have replaced the pair-wise comparison tests with interaction tests by 2-way ANOVA (see R1#2 in Major Comments), we have removed the Mann-Whitney U-test from Figure 1A-C.

4- Line 43: I believe "dHPC" is a typing mistake, it should be "mPFC"

We have corrected the typo in the Discussion.

5- The authors have explored the essentiality of dHPC with inhibitory dreadds while they use NMDA lesioning for mPFC. What was the rationale for these choices and why not use the same method for both experiments?

We had performed the lesion experiment in mPFC and DREADD inhibition in dHPC because we previously found these techniques were effective in inhibiting both regions, respectively (Terranova et al., *Neuron*, 2022). Since we have replaced the lesion experiment with additional optogenetics experiments in the revised manuscript (see R2#4 in Major Comments below), we have removed this text and other text corresponding to the lesion experiment.

REVIEWERS' COMMENTS

Reviewer #1 (Remarks to the Author):

I thank the authors for having addressed my comments so carefully and constructively, and consider that all my concerns have been addressed appropriately. This will provide a valuable contribution to the field.

Reviewer #2 (Remarks to the Author):

The authors have responded very sincerely to all my comments and have successfully and carefully revised the manuscript. I share with the authors my pleasure at the improvements made to the paper.

The effect of ArcT on axons remains controversial in this field of research, but I will not pursue it further here because of the changes in animal behavior in this experimental paradigm.

The use of the terms "allocentric" and "egocentric" in this manuscript differs from their traditional use in classical psychology. I do not disapprove of the authors' use of the terms, but in light of much previous work, it may be better to clarify the authors' definitions in this manuscript.

Reviewer #3 (Remarks to the Author):

The authors did remarkable work that significantly improved the original manuscript. I have no remaining concerns.

Reviewer #4 (Remarks to the Author):

The authors have done a great job addressing our comments.

Dear Editors and Reviewers,

We thank you very much for your time and effort in the review of our revised manuscript, "Systems consolidation induces multiple memory engrams for a flexible recall strategy in observational fear memory in male mice" (NCOMMS-22-34631A). We appreciate your comments and suggestions, which have greatly strengthened our manuscript. We address the following Reviewer comments on our revised manuscript as follows:

Reviewer #1 (Remarks to Author):

I thank the authors for having addressed my comments so carefully and constructively, and consider that all my concerns have been addressed appropriately. This will provide a valuable contribution to the field.

We thank the Reviewer for their support and constructive feedback.

Reviewer #2 (Remarks to the Author):

The authors have responded very sincerely to all my comments and have successfully and carefully revised the manuscript. I share with the authors my pleasure at the improvements made to the paper.

The effect of ArcT on axons remains controversial in this field of research, but I will not pursue it further here because of the changes in animal behavior in this experimental paradigm.

The use of the terms "allocentric" and "egocentric" in this manuscript differs from their traditional use in classical psychology. I do not disapprove of the authors' use of the terms, but in light of much previous work, it may be better to clarify the authors' definitions in this manuscript.

We thank the Reviewer for their support and constructive feedback. Because use of the terms "allocentric" and "egocentric" can be unclear, as pointed out by other Reviewers, we have removed these terms throughout the revised manuscript. Instead, we directly describe the experimental manipulations that we conduct for each experiment. Thank you for your feedback.

Reviewer #3 (Remarks to the Author):

The authors did remarkable work that significantly improved the original manuscript. I have no remaining concerns.

We thank the Reviewer for their support and constructive feedback.

Reviewer #4 (Remarks to the Author):

The authors have done a great job addressing our comments.

We thank the Reviewer for their support and constructive feedback.